# An instructive role for Interleukin-7 receptor α in the development of human B-cell precursor leukemia

Ifat Geron[1,2,3,15], Angela Maria Savino[1,2,3,15], Hila Fishman[1,2,3], Noa Tal[1,2], John Brown[4], Virginia A. Turati[4], Chela James[4], Jolanda Sarno[5], Michal Hameiri-Grossman[3], Yu Nee Lee[1,6], Avigail Rein[1,2,3], Hillary Maniriho[1,3], Yehudit Birger[1,2,3], Anna Zemlyansky[3], Inna Muler[2], Kara L. Davis[5], Victoria Marcu-Malina[7], Nicole Mattson[8], Oren Parnas[9], Rabea Wagener[10], Ute Fischer[10], João T. Barata[11], Catriona H. M. Jamieson[12], Markus Müschen[8], Chun-Wei Chen[8], Arndt Borkhardt[10], Ilan Richard Kirsch[13], Arnon Nagler[1,14], Tariq Enver[4] & Shai Izraeli[1,2,3,8✉]

Kinase signaling fuels growth of B-cell precursor acute lymphoblastic leukemia (BCP-ALL). Yet its role in leukemia initiation is unclear and has not been shown in primary human hematopoietic cells. We previously described activating mutations in interleukin-7 receptor alpha (IL7RA) in poor-prognosis "ph-like" BCP-ALL. Here we show that expression of activated mutant IL7RA in human CD34$^+$ hematopoietic stem and progenitor cells induces a preleukemic state in transplanted immunodeficient NOD/LtSz-*scid IL2Rγ*$^{null}$ mice, characterized by persistence of self-renewing Pro-B cells with non-productive V(D)J gene rearrangements. Preleukemic CD34$^+$CD10$^{high}$CD19$^+$ cells evolve into BCP-ALL with spontaneously acquired Cyclin Dependent Kinase Inhibitor 2 A (*CDKN2A*) deletions, as commonly observed in primary human BCP-ALL. CRISPR mediated gene silencing of *CDKN2A* in primary human CD34$^+$ cells transduced with activated IL7RA results in robust development of BCP-ALLs in vivo. Thus, we demonstrate that constitutive activation of IL7RA can initiate preleukemia in primary human hematopoietic progenitors and cooperates with CDKN2A silencing in progression into BCP-ALL.

[1] Felsenstein Medical Research Center and The Molecular Genetics and Biochemistry Department, Sackler Faculty of Medicine, Tel Aviv University, Petach Tikva, Israel. [2] Institute of Pediatric Research, Edmond and Lily Safra Children's Hospital, Chaim Sheba Medical Center, Tel Hashomer, Israel. [3] The Rina Zaizov Pediatric Hematology and Oncology Division Schneider Children's Medical Center of Israel, Petach Tikva, Israel. [4] Department of Cancer Biology, UCL Cancer Institute, UCL, London, UK. [5] Department of Pediatrics, Bass Center for Childhood Cancer and Blood Disorders, Stanford University, Stanford, CA, USA. [6] Pediatric Department and the Immunology Service, Jeffrey Modell Foundation Center, Edmond and Lily Safra Children's Hospital Sheba Medical Center, Tel-Hashomer, Israel. [7] Cytogenetic Unit laboratory of Hematology, Chaim Sheba Medical Center Tel Hashomer, Tel Hashomer, Israel. [8] Department of Systems Biology, City of Hope Comprehensive Cancer Center, Monrovia, CA, USA. [9] The Concern Foundation Laboratories at the Lautenberg Center for immunology and Cancer Research, IMRIC, Hebrew University Faculty of Medicine, Jerusalem, Israel. [10] Department of Pediatric Oncology, Hematology and Clinical Immunology, University Children's Hospital, Medical Faculty, Heinrich-Heine-University Düsseldorf, Düsseldorf, Germany. [11] Instituto de Medicina Molecular, Faculdade de Medicina, Universidade de Lisboa, Lisboa, Portugal. [12] UC San Diego, Moores Cancer Center, Division of Regenerative Medicine, Department of Medicine and Sanford Stem Cell Clinical Center, Ja Jolla, CA, USA. [13] Adaptive Biotechnologies, Inc., Seattle, WA, USA. [14] Hematology Division BMT and Cord Blood Bank Chaim Sheba Medical Center Tel-Hashomer, Tel-Hashomer, Israel. [15] These authors contributed equally: Ifat Geron, Angela Maria Savino. ✉email: sizraeli@gmail.com

The current paradigm of the evolution of B-cell precursor acute lymphoblastic leukemia (BCP-ALL) suggests two distinct stages: A commonly occurring initiating genetic event that generates preleukemia and rare progression to leukemia through the acquisition of additional somatic genetic events[1]. In childhood ALL, the initiating event occurs in utero and consists usually of an aberration in a transcriptional regulator[2]. Progression to leukemia is caused by a series of acquired genetic aberrations that halt B-cell differentiation and increase cell proliferation, survival and self-renewal[3–5]. Increased signaling through RAS or JAK-STAT pathways are typical progression events and are generally thought to act as the "fuel" enhancing leukemic cell growth[6,7]. Whether activation of signaling may have an instructive role affecting B-cell differentiation and initiating BCP-ALL is unknown and has never been demonstrated experimentally in human hematopoietic progenitor cells.

Interleukin-7 receptor alpha (IL7RA) is a receptor subunit with dual roles. Upon association with the interleukin-2 receptor gamma (IL2Rγ) subunit, it forms the Interleukin-7 (IL7) receptor and when bound to cytokine receptor-like factor 2 (CRLF2) subunit, it constitutes the thymic stromal lymphopoietin (TSLP) receptor[8,9]. Loss-of-function mutations in IL7RA are associated with absent B cells and T cells in mice but with the absence of only T cells in humans[10]. Thus, while IL7RA is important for mouse T and B lymphopoiesis its role in human B-cell development is unclear[11–13].

"Ph-like" leukemia is a subgroup of high-risk BCP-ALLs caused by activation of signaling leading to a similar gene expression signature to BCR-ABL1 ("Philadelphia") ALL[14–16]. The majority of these leukemias are characterized by aberrant expression of CRLF2/IL7RA and mutations activating JAK-STAT signaling[17]. We previously described IL7RA activating mutations in Ph-like BCP-ALL[18]. These mutations often introduce cysteine into the juxtamembrane domain resulting in dimerization of the receptor and constitutive signaling. Although attempted in the mouse, it is unclear if activation of IL7RA can initiate human BCP-ALL[19]. Moreover, the relevance of mouse modeling to human BCP-ALL is unclear, due to major differences in the role of both IL7 and TSLP signaling in B-cell development in mice and humans.

Here, we provide an experimental evidence in human hematopoietic cells that expression of activated IL7RA (IL7RAins) has an instructive role in human B-cell development by initiating a preleukemic state that is vulnerable to evolve to overt "Ph-like" BCP-ALL. We further demonstrate that the loss of cyclin-dependent kinase inhibitor 2A (CDKN2A) cooperates with IL7RA in the development of BCP-ALL.

## Results

**Activation of IL7RA pathway blocks differentiation of human B cells at the progenitor stage**. To test the role of activated IL7RA in leukemia initiation, we expressed wild type and/or an activated mutant form of human IL7RA containing an in-frame insertion (PPCL—p.Leu243_Thr244insProProCysLeu) (IL7RAins)[18] in human umbilical cord blood (CB) hematopoietic progenitors. As IL7RA mutations in BCP-ALL frequently correlate with aberrant CRLF2 expression, combinations of IL7RA and CRLF2 were used. The coding sequences were cloned into a lentiviral vector with a bi-cistronic cassette under the expression control of an Eμ-B29 promoter/enhancer to augment expression in B-cell precursors[20]. Backbone vector-expressing GFP (BB) was used as a control. Transduced CB hematopoietic progenitors (CD34+) were transplanted into NOD/LtSz-scid IL2Rγ^null (NSG) mice. (Supplementary Fig. 1 and Supplementary Table 1). The activity of the IL7RA/CRLF2 transgenes was verified by STAT5 phosphorylation assay in a BCP-ALL cell line (Supplementary Fig. 2).

The development of BCP-ALL leukemia is associated with a block in B-cell differentiation at pro/pre-B-cell stage[1,2]. We therefore analyzed the differentiation pattern of the human B-lineage cells 24–30 weeks post transplantation. B-cell differentiation beyond the pre-B-cell stage (CD19+CD10+sIgM−) was significantly inhibited in IL7RAins-transduced cells with or without CRLF2 (Fig. 1a, b and Supplementary Figs. 3 and 4a, b). To further define B-lineage differentiation stage of transduced cells, engrafted cells were analyzed by mass cytometry, and a single-cell developmental classifier was applied as previously described[21] (Supplementary Fig. 5). As depicted in Fig. 1c, an enlarged pre-BI population is observed in the CRLF2-IL7RAins-transduced cells and an earlier pro-BII fraction in cells transduced with IL7RAins alone.

Since the stage of B-cell differentiation is reflected by the V(D)J recombination status, we performed B-cell repertoire sequencing of the IgH locus in sorted BB and CRLF2-IL7RAins-transduced CD10+CD19+ cells from bone marrow (BM) of transplanted mice. Consistent with the early B-cell differentiation stage that was observed by immunophenotyping, the fraction of DJ rearranged cells was significantly expanded in the IL7RA-activated population (Fig. 1d). Activation of the IL7 pathway was previously reported to influence V(D)J rearrangements in lymphoid progenitors[22,23]. Indeed, sequencing of the IGH locus revealed that CRLF2-IL7RAins-transduced cells display increased N-nucleotide insertions during IgH rearrangements (Supplementary Fig. 4c), alluding to enhanced TdT activity. The relative block in B-cell differentiation was general as there was no statistically significant oligoclonal expansion (Supplementary Fig. 4d).

During normal B-cell differentiation, cells carrying nonproductive V(D)J rearrangements undergo programmed cell death[24]. In contrast, acute lymphoblastic leukemia cells often carry nonproductive V(D)J rearrangements[25]. We observed a substantial increase in the ratio of the nonproductive rearranged fraction in the CRLF2-IL7RAins-transduced cells (Fig. 1e). This observation suggests that signaling activation via TSLP/IL7RA signaling provided an enhanced survival capacity of the cells that would otherwise be destined to programmed death in the absence of productive B-cell receptor rearrangements.

**Aberrant expression of activated IL7RA induces a B-cell precursor population that retains self-renewal capacity**. B-cell precursor leukemic cells express the hematopoietic progenitor marker CD34 that is normally silenced past the early B-cell progenitor differentiation stage[26]. Consistent with the enrichment in early B-cell progenitors (Fig. 1c), activation of IL7RA pathway resulted in the expansion of the CD19+CD10+CD34+ population (Fig. 2a). In six out of 30 mice engrafted with activated IL7RA-transduced cells (with and without CRLF2), we have identified a unique CD10^high CD19+ subpopulation that was undetectable in control groups (Fig. 2b, Supplementary Fig. 6, and Supplementary Table 1 marked "p"). This population was enriched with CD34+ cells and can either represent expanded early B precursors or a population with a preleukemic potential.

One of the hallmarks of leukemic cells is the capability of self-renewal, a property of stem cells. To test whether expression of activated IL7RA affects self-renewal we re-transplanted 100,000–150,000 transduced cells that were harvested from the BM of primary mice 28–32 weeks after transplantation (Supplementary Fig. 1). As portrayed in Fig. 2c, the repopulation capacity of cells harvested from IL7RA transplanted primary mice was enhanced when compared to cells from mice that were transplanted with BB-transduced control. This was particularly evident in group transduced with the activated-IL7RAins only (P = 0.0073, ANOVA test). Thus both the differentiation arrest

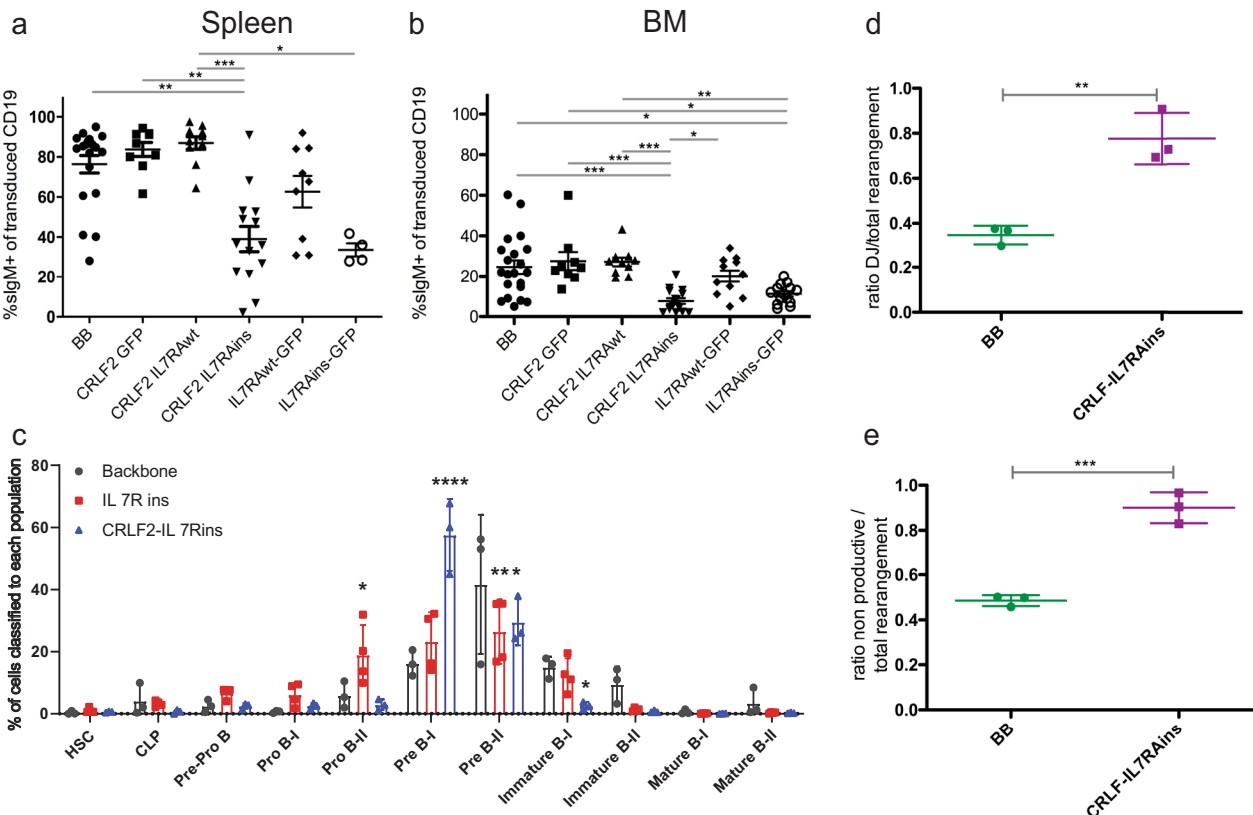

**Fig. 1 CRLF2/IL7RA transduction alters B-lineage differentiation of human CB CD34+ progenitors transplanted in immune-deficient mice. a** B-lineage differentiation to immature/naive B cells (sIgM +) of human CB CD34+ cells from the spleen (**a**) and BM (**b**) of engrafted mice expressing GFP (BB) [$n = 20$ (**a**), $n = 21$ (**b**)], CRLF2-GFP [$n = 9$ (**a**), $n = 9$ (**b**)], CRLF2-IL7RAwt [$n = 10$ (**a**), $n = 10$ (**b**)], CRLF2-IL7RAins [$n = 14$ (**a**), $n = 16$ (**b**)], IL7RAwt-GFP [$n = 9$ (**a**), $n = 11$ (**b**)] and IL7RAins-GFP [$n = 4$ (**a**) $n = 14$ 9 (**b**)]. Dot plots show sample scatter with mean +/− SEM. Each dot represents analysis of single mouse. Statistical analyses were performed using Kruskal–Wallis nonparametric test (Gaussian approximation) (**a**) $P < 0.0001$ Kruskal–Wallis statistic = 31.8 (**b**) $P < 0.0001$ Kruskal–Wallis statistic = 38.04. Gray linkers indicate statistically significant difference (*$P < 0.05$, **$P < 0.01$, ***$P < 0.001$) between groups in Dunn's post hoc analysis significance level α = 0.05. Gating strategy Supplementary Fig. 3. **c** Mass cytometery analysis of human cells from BM of engrafted mice. Points represent mean values (BB $n = 3$, CRLF2-IL7RAins $n = 3$, IL7RAins $n = 4$). Statistical analysis was done by two-way ANOVA followed by Tukey test for multiple comparison corrections. Individual variances were computed for each B-cell subgroup with CI of 95% (α = 0.05). The asterisks indicate statistically significant difference compared with the backbone group (*$P < 0.05$, **$P < 0.01$, ****$P < 0.0001$). Definition of each B-cell subgroup is detailed in Supplementary Fig. 4. **d**, **e** V(D)J rearrangement analysis of CD10+ and CD19+ BB/CRLF2-IL7RAins-transduced cells sorted from BM of transplanted mice. Each dot represent a value of a single sample. lines represent mean +/− SEM of BB ($n = 3$) and CRLF2-IL7RAins ($n = 3$). Gray linkers indicate statistically significant difference (*$P < 0.05$, **$P < 0.01$, ***$P < 0.001$) between groups. Statistical analyses were performed using two-tailed $t$ test. **d** Dot plot representing fraction of DJ rearranged of the total rearranged IgH loci in transduced cells $P = 0.0035$. **e** Dot plot representing ratio of nonproductive to total rearrangement in transduced cells $P = 0.0006$. Source data are provided as a Source Data file.

and the enhanced self-renewal are consistent with characteristics of preleukemic cells.

**Initiation of de-novo leukemia after serial transplantation of activated IL7RA hematopoietic progenitors.** BCP-ALL is the end result of sequential cumulative mutational events in which an initiating mutation causing "pre-leukemic state" is followed by secondary mutations that mediate progression to overt malignancy[2,4]. Transformation from preleukemia to leukemia in children is rare and often associated with an intervening period of several years. In agreement with this, none of the primary recipients of the transduced human hematopoietic progenitors developed leukemia within the first half-year of follow-up. This notwithstanding, we hypothesized that a selective pressure of serial transplantation might promote the evolution of pre-leukemic cells into leukemia.

Indeed, as depicted in Fig. 3, one out of the six IL7RAins-transduced CB that developed a clear CD10high CD19+ population in the primary transplanted mouse (bottom right sample in Fig. 2b) progressed to leukemia in a secondary transplanted

mouse. This leukemia was characterized by expansion of CD34+CD10+CD19+ population (Fig. 3a and Supplementary Fig. 7). To validate that the human-engrafted cells represented overt leukemia, tertiary transplants were performed in which all nine recipients developed identical leukemia within 8–15 weeks of transplantation (Supplementary Fig. 7).

VH-region sequencing of genomic DNA revealed that the leukemia was clonal and carried a nonfunctional (containing a stop codon) V3-15J4 gene rearrangement (Fig. 3b and Supplementary Fig. 8). Only this one allele was rearranged in the leukemic cells in agreement with a block of differentiation in an early B-cell stage. This was further supported by mass cytometry analysis classifying the leukemic cells as pro-B-II population (Supplementary Fig. 9).

Of note, repertoire sequencing of CD45+CD19+CD10+ cells from the primary mouse that generated leukemia detected the leukemic clone at a frequency of 0.09–0.02% of the CD10high CD19+ and CD10med CD19+ populations respectively (Supplementary Fig. 10). Thus, the leukemia was derived from a preleukemic clone in the primary transplanted mouse.

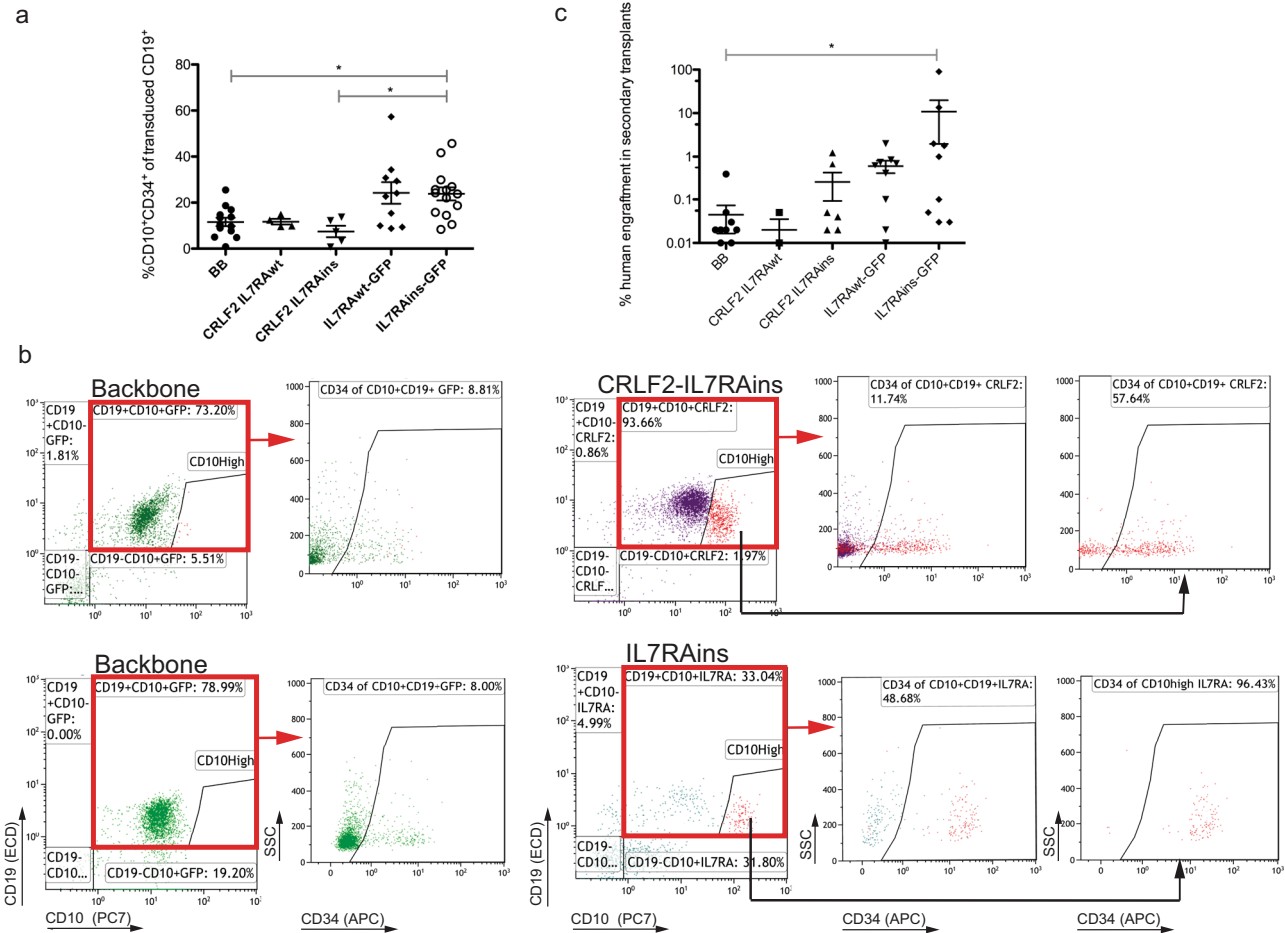

**Fig. 2 Enhanced CD34+CD10+ expression and self-renewal of IL7RA-activated cells. a** Relative CD10+CD34+ population of engrafted transduced human CD19+ cells in BM (BB $n = 11$, CRLF2-IL7RAwt $n = 4$, CRLF2-IL7RAins $n = 5$, IL7RAwt-GFP $n = 10$, IL7RAins-GFP $n = 14$). **b** Flow cytometry immunophenotyping of engrafted backbone and CRLF2-IL7RAins or IL7RAins-transduced cells. Samples in the same row are from the same CB batch. Arrows indicate that the gated population was analyzed in the following scatters. Gating strategy for (**a**), (**b**) in Supplementary Fig. 3. **c** Percentage of human cells in BM of secondary recipient mice that were transplanted with BM cells of primary engrafted mice (BB $n = 13$, CRLF2-IL7RAwt $n = 3$, CRLF2-IL7RAins $n = 8$, IL7RAwt-GFP $n = 9$, IL7RAins-GFP $n = 10$—six points with a value of zero were out of the logarithmic graph range, but were included in the mean and SEM calculations). **a**, **c** Dot plots depict sample scatter with mean $+/-$ SEM. Each dot represents the analysis of a single mouse. Gray linkers indicate a statistically significant difference (*$P < 0.05$) between groups. Statistical analyses were performed using Kruskal–Wallis nonparametric one-way ANOVA test with Dunn's post hoc analysis. Source data are provided as a Source Data file.

Karyotypic analysis and Cytoscan HD DNA array of the leukemic cells revealed several DNA copy number abnormalities (Fig. 3c and Supplementary Fig. 11). Of special interest is the deletion in 9p, with a biallelic deletion in CDKN2A, well-characterized progression events in the evolution of BCP-ALL[3,27] (Fig. 3c and Supplementary Fig. 12). Genomic analysis also detected PAX5 deletion (Fig. 3c) and an internal IKZF1 deletion (Supplementary Fig. 13), other typical progression events in high-risk Philadelphia and Ph-like BCP-ALL[3,28,29]. Importantly, none of these abnormalities were detected in cells from the same CB batch transduced with the control backbone vector, confirming that they did not exist in the germline prior to the transduction with the activated-IL7RAins receptor (Fig. 3c and Supplementary Figs. 12 and 13). The co-existence of PAX5, CDKN2A, and IKZF1 mutations classifies this leukemia as "Ikaros plus" high-risk ALL[30]. To better map the genetic landscape of the malignancy, we performed whole-genome sequencing (WGS, physical depth of 60×) of the leukemia sample and matched engrafted cells from the same CB transduced with BB. As portrayed in Supplementary Table 2, seven genomic deletions outside the immunoglobulin loci were evident in the leukemic

cells. This finding agrees with what was previously reported for *ETV6-RUNX1* human ALL and is hypothesized to be caused by the prolonged expression of the RAG recombination enzymes in the preleukemic and leukemic precursors[31]. In addition, pSTAT5 analysis by flow cytometry demonstrated a Ph-like typical constitutive activation of the JAK-STAT pathway that was cytokine-independent (Supplementary Fig. 14). Thus, experimental leukemia initiated by expression of IL7Rins in primary human hematopoietic progenitors recapitulated all the hallmarks of this type of ALL observed in patients.

**Single-cell analysis of B-cell precursor cells transduced with activated IL7RA reveals a distinct population with a strong Ph-like pro-survival gene signature**. To better characterize the genetic changes that preceded full leukemic transformation, we sorted human CD10+CD19+ cells from the BM of the leukemic mouse (Leukemia), the preleukemic mouse ("preleukemia CD10+CD19+"), and the mouse engrafted with matching CB transduced with control BB virus (BB control). In addition, we separately sorted the CD10highCD19+ ("preleukemia CD10high" see experiment illustration Fig. 4a) subpopulation presumably

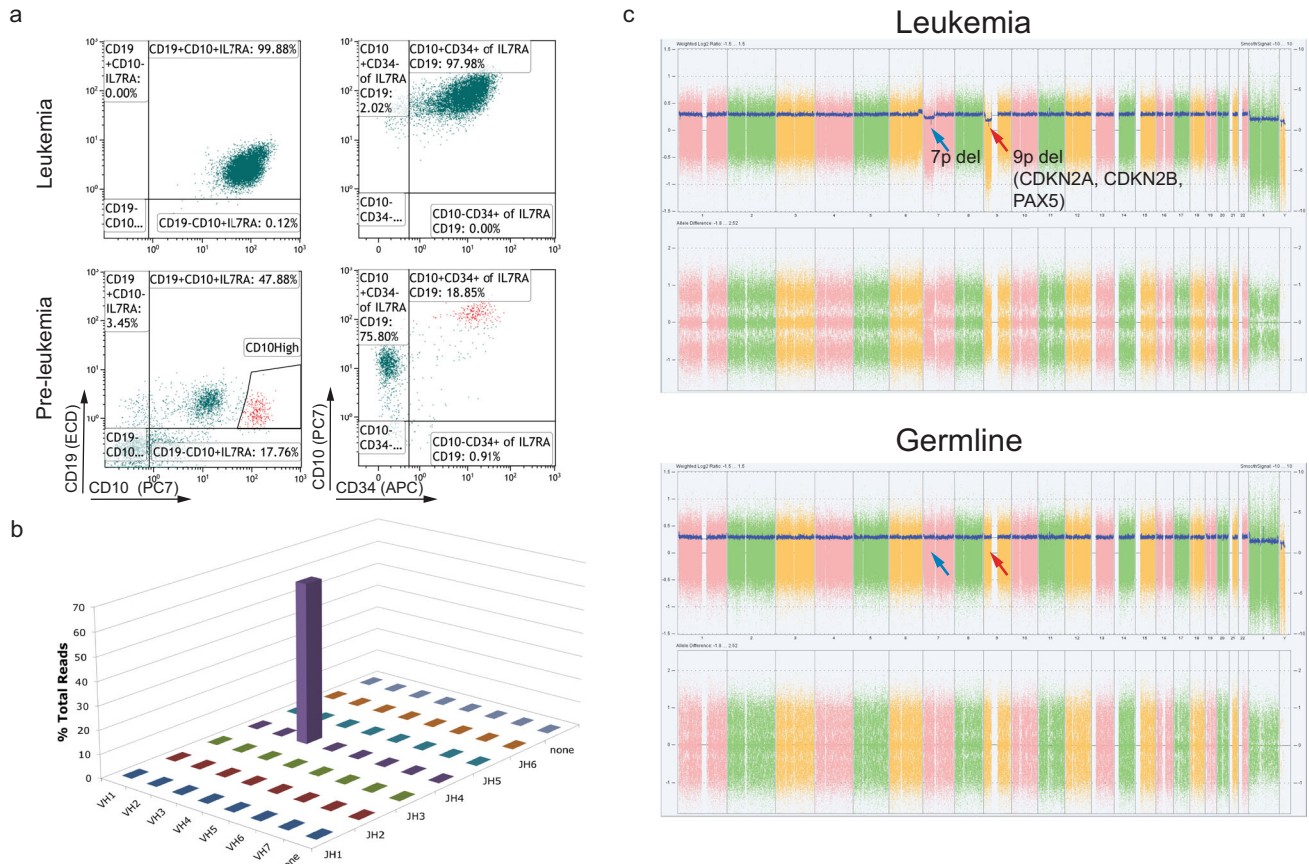

**Fig. 3 Secondary transplantation of IL7RA-activated human hematopoietic progenitors result in the development of clonal B-cell precursor leukemia. a** Flow cytometer scatter plot of human-engrafted cells in BM of the leukemic mouse. **b** Bar-graph of V–J rearrangements in leukemic population. The bars represent counts in the sequenced library of B-cell receptor rearrangements. Source data are provided as a Source Data file. **c** Genomic SNP array analysis of leukemic cells (Leukemia) and of BB-transduced engrafted cells from the corresponding cord blood (representing germline).

enriched with preleukemic cells because of their high resemblance to the leukemic immunophenotype and higher abundance of leukemic VDJ rearranged clone (Supplementary Fig. 10). Sorted cells were subjected to 10× single-cell gene expression analysis.

T-distributed somatic neighbor embedding (T-SNE) plot demonstrated a clear separation between three cell populations: the leukemic cells, the BB control cells, and the CD10high cells from the preleukemic mouse (Fig. 4b). The total CD10+CD19+ population of the preleukemic mouse was distributed between the CD10high cluster and the BB control cluster.

Differential expression analysis of the populations (see differential gene list in Supplementary File DEGs.xls) shows a close hierarchical relationship between the CD10+CD19+ (without the CD10high population) from the preleukemic mouse cells and the control backbone-transduced cells (only 57 differentially expressed genes—Fig. 4c). In contrast, comparison of the leukemic cells and the CD10high cells from the preleukemic mouse to the BB-transduced cells revealed 293 and 432 differentially expressed genes, respectively (Fig. 4c). In total, 152 of these differentially expressed genes were shared between the leukemia and CD10high groups, thus supporting the hypothesis that the CD10highCD19+ compartment is enriched with preleukemic cells. Functional analysis (https://david.ncifcrf.gov[32]) demonstrated activation of B-cell signaling pathways both in the leukemic and the CD10highCD19+ "preleukemic" cells (Supplementary Fig. 15A). In addition, the CD10highCD19+ cluster was enriched with cell cycle pathway genes (Supplementary Fig. 15A, B).

In addition to the scRNAseq, we also performed bulk RNAseq of transduced CD19+ CRLF2-IL7RAins (IL7RA-activated) and BB (BB control) cells that were sorted from five matched-CB-batches transplanted mice. RNAseq analysis of the IL7RA-activated cells revealed, as anticipated for activated CRLF2/IL7RA signaling[33], enrichment of gene sets representing JAK-STAT and mTOR signaling (Supplementary Fig. 16). Consistent with the phenotypic B-cell differentiation block, the signatures were enriched with B-cell precursor gene expression and higher expression of RAG1 and RAG2 compared to the BB-transduced group (Supplementary Fig. 16 and Supplementary Data 1 and 2). Elevated levels of RAG1 (but not RAG2) transcript were detected by scRNAseq both in the preleukemic and in the leukemic populations (Fig. 4d). We speculate that similar to what was recently described during V(D)J recombination[34] and specifically reported for ETV6-RUNX1 ALL[31], increased RAG1/2 activity in B-cell precursor preleukemia might lead to genetic instability. Consistent with the expanded N-nucleotide insertions in B-cell precursors expressing activated IL7RA (Supplementary Fig. 4d), the expression of DNTT encoding the TdT enzyme was elevated in both the preleukemia CD10high and leukemic cells (Fig. 4d). Interestingly, the expression level of PAX5 was not altered in the leukemic cells, and somatic mutations were not found in the coding region of the remaining allele by Sanger sequencing of the cDNA (Fig. 4d).

To investigate how closely the experimental leukemia recapitulated primary human Ph-like ALLs, we compiled ranked lists of the differentially expressed genes between the leukemia and

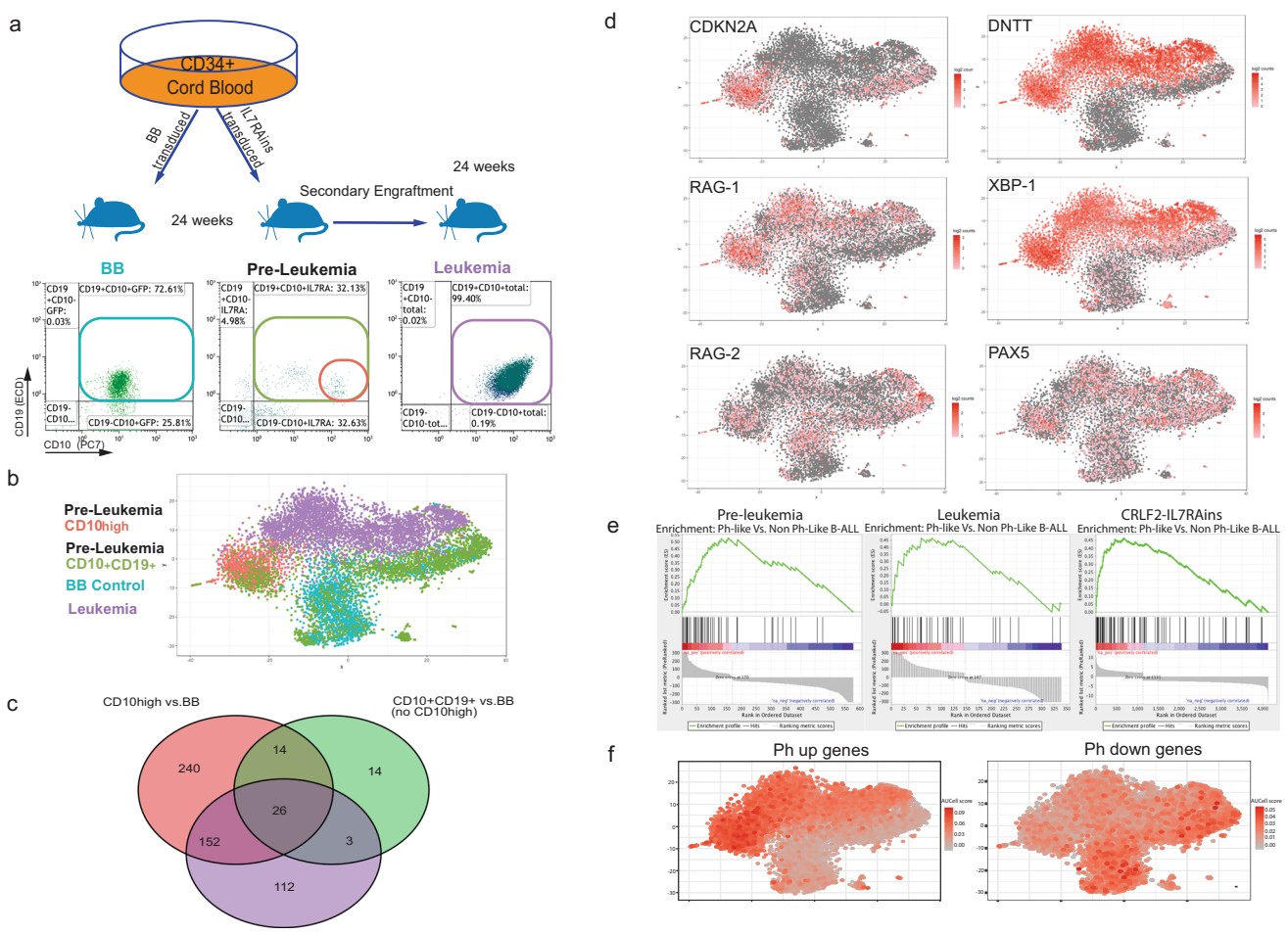

**Fig. 4 Philadelphia-like ALL gene signature in bulk and scRNAseq analyses of activated IL7RA-engrafted cells. a** Scheme of sample acquisition for scRNAseq. **b** Transcriptome correlation t-SNE map after 10X scRNAseq. **c** Venn diagram of differentially expressed genes in bulk analysis of CD10+CD19+ and CD19+CD10high cells of the preleukemic mouse and leukemia sample vs BB control sample Source data are provided as a Source Data file. **d** Relative expression of selected genes displayed on t-SNE map. **e** GSEA enrichment plot of preleukemic cells, leukemic cells and CRLF2-IL7RAins over BB differentially expressed genes aligned to Ph-like vs non Ph-like ranked gene list (**f**) AUCell analysis for single-cell expression of Ph-like (Ph) gene set projected on t-SNE map.

CD10high cells from the preleukemic mouse and control BB cells and used it in gene set enrichment analysis against list of differentially expressed genes from two groups of BCP-ALL: Philadelphia and Ph-like BCP-ALL versus combined groups of BCP-ALL leukemias (Patient database St. Jude's group-GSE26281 https://www.ncbi.nlm.nih.gov/geo/query/acc.cgi). As seen in Fig. 4e, a Philadelphia-like gene signature was found both in the leukemic and CD10high samples when compared to the BB control (Philadelphia gene set is provided in Supplementary Data 3, Supplementary material file "Ph geneset" also see major genes contributing to enrichment in Supplementary Data 4). Similar results were obtained from analysis of bulk RNAseq of activated IL7RA vs BB control (Fig. 4e and Supplementary Data 4).

To better map the cells with Ph-like signature, area under curve for single cells (AUCell) analysis[35] was preformed using the above list. As can be seen in Fig. 4f, expression of the Ph-like upregulated genes was high in the leukemia and CD10high clusters while the score of Ph-like downregulated genes of the gene set was higher in the BB and CD10+CD19+ clusters.

Furthermore, to test the clinical relevance of our experimental model, we assessed the expression of 15 genes that were

developed for clinical use in diagnosis of Ph-Like patients[36,37]. Strikingly, 12 genes from this list out of 13 that were detected in the RNAseq analysis, were predominantly expressed in the CD10high cells and in the leukemic cells whereas only one (*PON2*), did not show a preferential expression pattern (Supplementary Figs. 17 and 18).

In agreement with their aberrant survival at the presence of nonproductive immunoglobulin heavy chain gene V(D)J gene rearrangements, IL7RA-activated cells also displayed significant enrichment for gene sets associated with survival/proliferation pathways (MYC pathway, rescue of apoptosis by IL6 signaling—Supplementary Fig. 16 and Supplementary Data 2). Furthermore, enrichment of the unfolded protein response gene set, with upregulation of XBP1 (FC 1.95) and HSPA5 (FC 1.42) that were previously demonstrated to be essential for pre-B and pre-B-ALL cells survival[38] were detected in the IL7RA-activated and in the leukemia and preleukemia CD10high groups, suggesting a role for early unfolded protein response in the transformation (Fig. 4d, Supplementary Fig. 16, and Supplementary Data 1 and 2). Thus, gene expression analysis demonstrated activation of pathways promoting the survival of transduced B-cell progenitors, comparable to human Ph-like BCP-ALL.

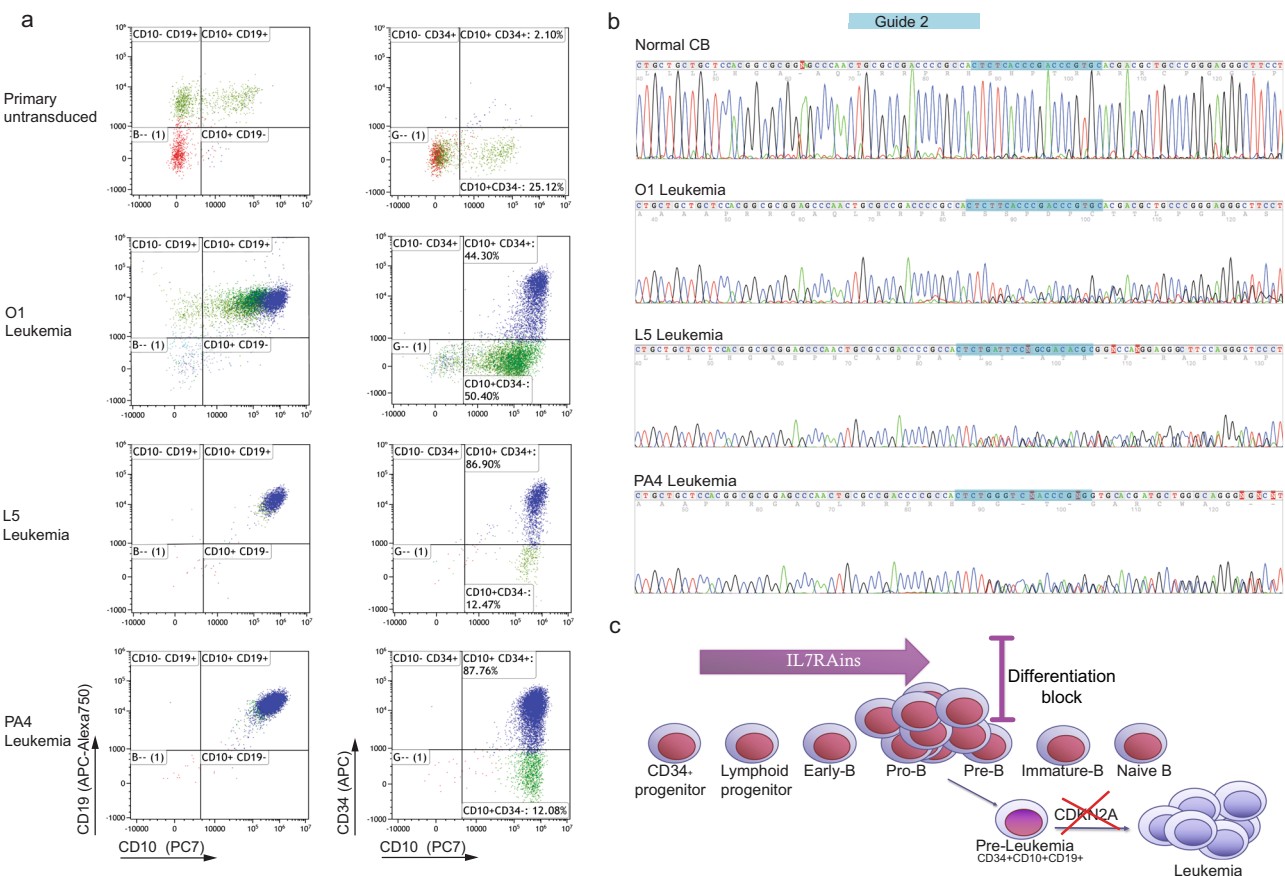

**Fig. 5 CDKN2A disruption cooperates with IL7Rins for full leukemic transformation CD34 + human cord blood (CB) progenitors. a** Flow cytometer scatter plot of human-engrafted cells in BM of untransduced CB and leukemic mice. **b** Sanger sequencing electropherogram of gDNA from CD45+ cells of leukemic mice surrounding guide 2 targeting CDKN2A, demonstrating disruption of the locus in leukemic cells. **c** Scheme of leukemia development after aberrant activation of IL7RA.

**CDKN2A disruption cooperates with activated IL7RA in leukemic transformation**. The tumor suppressor CDKN2A was significantly upregulated in the IL7RA-activated cells of the pre-leukemic mouse, as seen in the scRNAseq (Fig. 4d) as well as in the bulk RNAseq (Supplementary Data 1 and 2). Consistent with its biallelic deletion in the leukemic cells (Fig. 3c and Supplementary Fig. 12) it was not expressed in the leukemic cells (Fig. 4d). Loss of *CDKN2A* is universally observed in primary human BCP-ALL with mutated activated IL7RA[3]. Together, these observations suggest, that enhanced proliferation of the self-renewing B-cell precursors expressing the mutant IL7RA is counteracted by the expression of the cell cycle "gatekeeper" proteins coded by CDKN2A. The loss of the negative cell cycle regulator CDKN2A may be the trigger for the evolution of IL7Rins preleukemic cells to fully transformed cells.

To test this hypothesis, we targeted CDKN2A with CRISPR-CAS9 (gCDKN2A) lentivirus in CD34+CB cells. Two guides were selected to target both CDKN2A gene products p16 (INK4A) and p14 (ARF). A mix of the two gCDKN2A viruses was used to transduce the cells. As a control, we used CRISPR-CAS9 lentivirus with a guide targeting Luciferase sequence (gLuciferase) (Supplementary Fig. 19 and Supplementary Table 3). gCDKN2A and gLuciferase-transduced cells (sorted GFP+) were re-transduced with IL7Rins or backbone lentivirus (gCDKN2A-IL7Rins/gCDKN2A-BB, gLuciferase-IL7Rins/gLuciferase-BB) and transplanted in NSG mice (Supplementary Fig. 19).

Of seven CB batches independently transduced with gCDKN2A-IL7Rins, three evolved to leukemia after transplant

in NSG mice [Two (O1 and L5—Fig. 5a) after 18 and 22 weeks, respectively, in primary mice and one after 14 and 20 weeks in two different secondary mice (PA4, PA2 Fig. 5a and Supplementary Fig. 20). No leukemia developed from cells that were transduced with gLuciferase-IL7Rins (from six CB batches) or the targeting vector alone in the gCDKN2A-BB group (seven CB batches). The leukemias were CD34+CD10[high]CD19+ positive (Fig. 5a) with V(D)J clonal markers (Supplementary Table 4) and repopulated secondary transplanted mice (Supplementary Fig. 20).

We confirmed that the leukemic cells harbored genetic editing that disrupt the *CDKN2A* locus (Fig. 5b and Supplementary Figs. 21 and 22). Importantly, sequencing of CDKN2A RNA that was isolated from the leukemic cells demonstrated out-of-frame insertion/deletions preventing protein expression (Supplementary Figs. 22 and 23). The editing of CDKN2A in addition to the IL7Rins mutant was also confirmed by exome sequencing of the three leukemias. One of the leukemias harbored additional deletions in PAX5 and IKZF1, thus can be classified as IKAROS plus (Supplementary Fig. 14) similar to the spontaneous leukemia that was developed after IL7Rins transduction. No additional pathogenic or likely-pathogenic variants were identified in the leukemias by exome sequencing. Genetic editing of CDKN2A was detected also in minor populations of cells transduced with backbone and targeting CDKN2A editing vectors (Supplementary Fig. 25) although the equivalent quantity of gCDKN2A transduced and sorted cells were transplanted in the backbone control group. Thus, the mere targeting of CDKN2A did not lead

to cell expansion and leukemic transformation. Together, these findings demonstrate that mutational activation of IL7RA cooperates with CDKN2A silencing in progression to leukemia.

## Discussion

Childhood ALL is preceded by a clinically silent phase of pre-leukemia detectable only through molecular genomic approaches. While the preleukemic state is estimated to be fairly common (up to 1:20 children[39]), transformation to leukemia is rare.

This observation places emphasis on understanding how the preleukemic phase is initiated and how the responsible lesions can pre-dispose cells for subsequent frank transformation. Herein, we explored this issue in the most common subtype of the poor-prognosis Philadelphia-like (Ph-like) B-cell precursor acute lymphoblastic leukemia (BCP-ALL). This variant is commonly associated with aberrant expression of cytokine receptor-like factor 2 (CRLF2) which dimerizes with Interleukin-7 receptor alpha (IL7RA) to form the receptor for Thymic Stromal Lymphopoietin (TSLP)[9]. Additional mutations in IL7RA itself further underscore the importance of the IL7RA receptor axis in Ph-like BCP-ALL[14,18]. Here, we wished to determine whether IL7RA signaling has any role in the initiation of human BCP-ALL and, if so, how the receptor activation-driven B-cell precursors are pre-disposed to transformation. Recent research described the ability of IL7RA activation to initiate B-cell precursor ALL[40,41] in mouse hematopoietic cells. However, due to the potential different functions of IL7 signaling in B-cell development in mice and humans and given the heterogeneous phenotypes of IL7RA perturbations in mice[8–11,42–44], we chose to establish an experimental model that authentically re-capitulate disease initiation and progression in relevant human cells.

We addressed that question by expression of activated IL7RA (IL7RAins) in human hematopoietic cells and showed that expression of activated IL7RA (IL7RAins), with or without CRLF2, modify human B-cell development into a state that could evolve into an overt "Ph-like" BCP-ALL. We further demonstrated the critical role of CDKN2A in IL7RA-activated B-cell progenitors for the progression to full-blown leukemia.

Interestingly, we did not observe the development of T-ALL, despite the important role of IL7RA in T-cell development and the presence of IL7Rins in T-ALL[8,45]. This may be due to the use of B-cell enhancer in our lentiviral vector[20] and the xenografting environment in the NSG strain that is highly permissive of human B cells[46].

The CD10$^+$CD19$^+$ compartment of activated IL7RA-transduced B-cell progenitors was enriched in early B-cell precursors with typical immunophenotype of BCP-ALL (CD34$^+$CD10$^+$CD19$^+$). These cells carried an increased frequency of loci with only DJ and/or nonproductive V(D)J rearrangements of the IgH chain. Hence, we contemplate that this population encompasses "pre-leukemic" cells. The survival of these cells, normally subject to apoptosis, might be explained by the increased expression of BCL2L1, previously described to rescue pro-B cells with aberrant V(D)J rearrangements from apoptosis[47]. The expression of other BCL2 family members and pro-survival genes and early unfolded protein response was also increased. Interestingly, we detected an augmented expression of the cell cycle inhibitor CDKN2A in the preleukemic cells, acting as a "gatekeeper" in restraining aberrant proliferation[48] and, possibly, extending the time of the cells in the B-cell progenitor stage. Molecularly, the transduced cells displayed the typical "Ph-like" leukemia gene expression signature. We also observed significantly elevated expression of RAG1/2 as well as DNTT encoding TdT in the IL7RA, activated cells. This is consistent with B cells that are held in an early progenitor stage and leads us

to speculate that genetic instability driven by prolonged RAG and TdT activity might, as previously demonstrated[31], be involved in the eventual development of leukemia, for example, by enabling deletions of CDKN2A and IKZF1.

A possible explanation to the early block of differentiation, observed as a result of constant activation of IL7RA, lays in the role played by IL7 signaling in mouse B-cell development; whereby downregulation of the pathway and switch to pre-BCR signaling, the expression of Bcl6 and of Ikaros, promotes normal progression of B-cell differentiation[49]. Thus, our observations in human B-cell progenitors resonate well with the earlier finding that in mouse cells constitutive activation of IL7 signaling results in cell differentiation arrest at an early B-cell stage[50]. Our observations suggest that IL7 role in human and mouse B-cell differentiation may be more similar than previously contemplated.

Single-cell RNAseq analysis suggested that preleukemic cells resided within a subpopulation of early B-cell precursors with CD34$^+$CD10$^{high}$CD19$^+$ immunophenotype[51]. This population harbored the specific leukemic V(D)J clone. The experimental leukemia presented with all the hallmarks of Ph-like "IKZF1 plus" human BCP-ALL in patients including a Ph-like expression signature, the spontaneous acquisition of the genomic loss of IKZF1, and the loss of the cell cycle regulator CDKN2A[30]. Hence, we successfully created an experimental model for IL7RA driven BCP-ALL.

The singularity of the leukemic event in mice that were transplanted with activated IL7RA CB alone with a maximal follow-up of less than a year suggest that the rare specific spontaneous genomic events, such as the biallelic loss of CDKN2A, are required for the cooperation with IL7Rins for progression to leukemia. Through gene editing of the CDKN2A locus in primary human hematopoietic progenitors, we have directly confirmed this hypothesis. Thus, we demonstrated here that activated receptor signaling can initiate preleukemic state that evolves to "Ph-like" BCP-ALL through the loss of CDKN2A (Fig. 5c).

## Methods

**Human CD34$^+$ hematopoietic progenitors**. Fresh cord blood (CB) units were obtained from Sheba Medical Center CB bank under Institutional Review Board-approved protocols to obtain CB units for research purposes (Approval 5638-08-SMC), Donations of cord blood to public cord blood bank are recruited among labors in the obstetric delivery department. Informed consent is signed that cord blood specimens that are not suitable for banking will be used for research. No compensation is granted upon donation. CD34$^+$ cells were isolated using magnetic beads (Miltenyi, USA) following the conventional method.

**IL7RA and CRLF2 ectopic expression**. IL7RA and CRLF2 were cloned from previously cloned cDNA[18,33] into B-cell-specific lentiviral vectors (Kindly provided by Rawlings Lab[20]) with standard cloning protocols.

**CRISPR CAS9 targeting of CDKN2A**. CRISPR guide sequences targeting the coding region of CDKN2A were designed using the Genetic Perturbation Platform (Broad Institute)[52]. Guide1:ACGCACCGAATAGTTACGGT (located at chr9:21974693 to 21974712 GRCh38/hg38) Guide2: GTGCACGGGTCGGGTGAGAG (located at chr9:21971110 to 21971129 GRCh38/hg38) control Guide (targeting Luciferase): GATTCTAAAACGGATTACCA. Oligonucleotides (forward and reverse strands) for each sgRNA were annealed into a double-strand DNA fragment with 5′-CACC and GTTT-3′ overhangs and cloned into the Lenti-CRISPRv2GFP (Addgene) lentiviral vector (hU6-driven sgRNA co-expressed with EF-1 α-driven Cas9 endonuclease and green fluorescent protein [GFP]) using the BsmBI (NEB) restriction sites. Each construct was validated by Sanger sequencing using hU6-F primer 5′-GAGGGCCTATTTCCCATGATT-3′. Lentivirus was produced in HEK293 cells (ATCC) by co-transfecting sgRNA constructs with the packaging plasmids pPAX2 (Addgene) and pMD2.G (Addgene).

**Virus production and titer**. Production of lentiviral vectors was done according to Tiscornia, G[53]: 3rd generation lentivector packaging plasmids for pRRL-Eµ-B29 vectors and 2nd generation lentivector packaging plasmids for LentiCRISPRv2GFP were co-transfected into 293T cells in the ratio of (15:10:5:4) (Lenti-vector:pMDL:pVSVG:pREV) and 2.8:2:1 (Lentivector pSPAX: pMD2G)

respectively using ProFection Calcium Phosphate mammalian transfection system (Promega) or lipofectamine 30000 (Thermo) according to manufacturer's protocol. Transfection medium was replaced 6–15 h after transfection with 5% serum DMEM serum and virus-containing supernatant was collected 24 and 48 h after replacement. The supernatant was then filtered with 0.45-μm PVDF filters (Millipore, MA, USA) and centrifuged in ultra-centrifuge using SW28 rotor for two and a half hours in 70,000 × g. The virus was reconstituted in 300–600 SFEM medium (STEMCELL Technologies, Vancouver, British Columbia, Canada). The concentrated virus was frozen at −80 °C until use. An aliquot of the frozen virus was used for titer in 018Z cells percentage of transduced cells was evaluated by flow cytometry using GFP, CRLF2, or IL7RA antibodies (Biolegend, CA, USA) Titer (infectious units/ml) was calculated according to Eq. (1):

Equation (1): titer calculation

$$\frac{\% \, transduced \, cells X \, \# \, cells \, at \, day \, of \, transduction}{total \, \mu l \, of \, virus/well} X1000 = virus IU/ml$$

**Transduction of CB CD34+ hematopoietic progenitors**. In total, $5 \times 10^4$–$7.5 \times 10^4$ CB CD34+ cells were plated in 96 U bottom-well plate (Corning Incorporated, NY, USA) in 50–100 μl SFEM (STEMCELL Technologies Vancouver, British Columbia, Canada) supplemented with hSCF (100 ng/ml) hFLT3 ligand (100 ng/ml) TPO (20 ng/μl) and IL6 (20 ng/μl). Cells were transduced by addition of virus in MOI of 50–200 and spin (800 × g 32 °C 45 min no break). Four to eight hours after spin, the wells were supplemented with fresh media. For single-virus transduction, cells were transduced twice in consecutive days. Prior to the second transduction, old media containing virus was discarded. For combination transduction, cells were transduced once with the first virus, sorted 48 h post transduction and positive cells were transduced with the second virus. Transduction efficiency was evaluated by flow cytometry using GFP or CRLF2/IL7RA antibodies. For single-virus transduction, all cells in transduction wells were collected and transplanted. For combination transduction, sorted cells after secondary transduction were supplemented with negative sorted cells (from the same condition i.e., GFP—from gCDKN2A or GFP—from gLucifarase were added to IL7RA/BB gCDKN2A or IL7RA/BB gLuciferase accordingly) for 50–150 K total (20–30 K transduced) cells per mouse. comparable numbers of sorted transduced cells were used per mouse from the same CB batch.

**Xenotransplants**. NOD/LtSz-scid IL2Rγnull (NSG) mice were purchased from Jackson laboratories (Mount Desert Island, Maine, USA). Mice were bred and housed in specific pathogen-free conditions. All animal experiments were approved by the Animal Care Committee at Sheba Medical Center (IRB 1007/15) or by the Animal Care Committee at Rabin Medical center (022_b15189 040419). In all, 5–8-week-old NSG females were irradiated (1–1.5 Gy X-ray) or treated with Busulfan (25 mg/kg) 4–24 h prior to transplantation. For primary transplantations, $1 \times 10^5$–$1.5 \times 10^5$ cells were transduced 72–96 h prior to transplantation with MOI of 50–200 by spinfection. On transplantation days cells were sampled to assess transduction efficiency and transplanted via tail vain injection. Mice were bleed once every 2–3 weeks starting 10 weeks post transplant to monitor engraftment. In all, 23–32 weeks post transplantation, or upon sign of disease (blasts in PB/lethargy/10% body weight loss) mice were euthanized and hematopoietic tissues (spleen, bone marrow from femurs, liver and peripheral blood) were harvested. In total, 250 mice were used in the described experiments, including primary and secondary/tertiary transplantations.

**Flow cytometry and sorting**. Standard staining protocols were used for the sort and analysis of cells. All antibodies that were used are specific to human antigens. In brief, cells were washed in staining media (2% FBS in PBS) and resuspended in staining media containing fluorochrome-conjugated antibodies, blocking antibodies when mouse tissue was used, and 7AAD for 30 min. (Supplementary Table 6). Following staining, cells were washed and analyzed on Gallios/Cytoflex S flow cytometer (Beckman-Coulter, CA, USA) or sorted using ARIA I/Aria III/Melody FACS sorter (BD Biosciences, San Jose, CA, USA). Single stains and FMOs (Fluorescent minus one staining) of each fluorophore were used for cytometer setup and gating. Analysis was performed using Kaluza software (Beckman-Coulter, CA, USA) on live cells after the exclusion of 7AAD-positive stained cells.

For xenografts sample analysis, hematopoietic tissues (spleen, bone marrow (BM), and peripheral blood (PB)) were harvested from mice at sacrifice time and kept throughout the processing time on ice. BM cells were flushed from the hind leg bones and strained through a 70-μm mesh cell strainer. Spleen and liver were mashed on a 70-μm mesh cell strainer. PB and spleen were subjected to red blood cell lysis (Biolegend, San Diego, CA, USA) per the manufacturer's protocol. Cells that were not used for analysis/sort were viably frozen in FBS + 10% DMSO.

For RNAseq and repertoire analysis, processed xenograft samples were stained as described above. For RNAseq, 5000–20,000 Live CD45+ CD3− CRLF2/GFP+ cells were sorted directly into mini-centrifuge tubes containing 800 μl of cold TRIzol (Thermo Fisher Scientific Waltham, MA USA). Tubes were vortexed immediately after sort and flash-frozen in liquid nitrogen for further RNA purification. For repertoire analysis, 5000–20,000 live CD45+ CRLF2/GFP+ CD10+ and CD19+ cells were sorted directly into mini-centrifuge tubes containing 200 μl STM. Cells were

then pelleted at 800 × g for 10 min and kept at −20 °C or processed immediately for gDNA extraction. For single-cell RNAseq 4000–10,000 cells were sorted.

For phosphorylation assays, cells (from subconfluent culture) were first washed and starved for 4 h (in media with no cytokines). Cells were then incubated with cytokines (hIL7, hTSLP) for 20 min, washed and stained with LIVE/DEAD fixable per the manufacturer's protocol (Thermo Fisher Scientific, Waltham, MA, USA (molecular probes brand)), cells were then stained for cell surface markers, fixed with 1.5% formaldehyde for 10 min, porated with ice-cold MeOH while vigorously vortexing and incubated at 4 °C for at least 10 min. Cells were then stored overnight or more (up to 2 weeks) at −20 °C. Fixed cells were then washed twice in staining media then resuspended in staining media containing pSTAT antibodies and re-stained for surface markers. Stained cells were analyzed on Gallios™ Flow Cytometer (Beckman-Coulter, CA, USA).

**Mass cytometry analysis**. Samples were processed as follows: Bone marrow samples (backbone ($n = 3$), CRLF2/IL7ins ($n = 3$), IL7ins ($n = 4$), and healthy BM ($n = 3$)) were thawed, stained with cisplatin to determine viability, rested for 30 min at 37 °C and then perturbed with IL7 (100 ng/mL) for 15 minutes (only for Leukemia and control BB BM) before being fixed with formaldehyde 1.6% for 10 min at room temperature. Cells were then barcoded using palladium-based labeling reagents, collected in one tube, stained with surface antibodies and after being permeabilized with methanol stained with intracellular Antibodies (Supplementary Table 5). Finally, cells were stained with 191/193Ir DNA intercalator before being analyzed the Helios mass cytometer (Fluidigm, Inc., South San Francisco, CA). Normalization of signal intensity loss during the CyTOF run was controlled utilizing metal standard beads mixed with the sample during the data acquisition.

Mass cytometry data were then analyzed using Cytobank (Cytobank Inc., Mountain View, CA) and were run through a B-cell developmental classifier recently described[21]. Specifically, healthy bone marrow (run with the samples) was manually gated into 11 consecutive developmental stages of B lymphopoiesis. The mean arsinh-transformed expression of ten markers (CD45, CD20, CD24, CD34, CD38, IgMi, TdT, CD19, IgMs, and CD10) was determined for each healthy population, and single cells from each sample were assigned to the most similar healthy population based on the shorted Mahalanobis distance calculated from the expression of the same ten markers.

**RNA/DNA sequencing and expression profile analysis**
*Bulk RNA sequencing*. Total RNA was purified from 5000 to 20,000 transduced (CRLF2+ or GFP+) CD45+CD3− cells that were sorted from spleens of transplanted mice. cDNA libraries were prepared using SMARTer v3 kit (Clontech Laboratories, Inc. A Takara Bio Company, Mountain View, CA, USA) followed by the Nextera-XT indexing protocols (Illumina, CA, USA). Genome-wide expression profiles were obtained by sequencing the samples on Illumina NextSeq 500 using NextSeq 500/550 High Output v2 kit.

*Bulk RNA differential expression analysis*. Paired expression analysis of CRLF2-IL7RAins versus BB was performed as follows: The Log₂ of the counts+1 was first calculated. The fold change (FC) of expression was defined as the differences between Log₂ CRLF2-IL7RAins counts and Log₂BB counts within the same cord blood batch. Significantly differential expressed genes were ranked per the average FC or their paired $t$ values calculated as described in Eq. (2).

Equation (2): t values

$$t = \frac{\bar{d} - 0}{S_d / \sqrt{n}}$$

when $\bar{d}$ = avarage FC, $n = 6$ pairs of samples and $S_d = \sqrt{\frac{1}{n-1} \left\{ \left( \sum_1^n d_i^2 \right) - n\bar{d}^2 \right\}}$.

Genes with overall low counts (background levels) in both samples (CRLF2-IL7RAins and BB) were filtered out by count sum <30. Significance of the result was determined by t value: $P < 0.1$ when $t(5,0.95) > 2.015$ and $P < 0.05$ when $t(5,0.975) > 2.571$. Ranking differential expressed genes by the significance of the change created lists of genes for further analyses:

*GSEA analysis*. GSEA algorithm was used as described[54] to evaluate enrichment of CRLF2-IL7RAins gene signatures in Philadelphia and Philadelphia-like cases compared to non-Philadelphia-like cases. Gene expression data for B-ALL patients was obtained from the patient database St. Jude's group (GSE26281 https://www.ncbi.nlm.nih.gov/geo/query/acc.cgi). This database included 29 Philadelphia and Philadelphia-like B-ALL cases [BCR-ABL ($n = 18$), CRLF2 + ($n = 11$)] and 98 non-Philadelphia-like B-ALL cases [E2A-PBX ($n = 8$), TEL-AML1 ($n = 24$), MLL rearrangements ($n = 15$), non CRLF2 + hyperdiploidy ($n = 29$), other ($n = 22$)]. Ranked list was generated using free GEO website tool GEO2R (see Supplementary Data ph geneset.xls).

*Single-cell RNA sequencing (scRNAseq)*. 10X library was prepared from 4000–10,000 cells (10X V3 library preparation kit, 10X, USA) that were sorted

from BM of leukemic/preleukemic/BB engrafted mouse and sequenced on NextSeq 500 System (Illumina, USA).

*Single-cell RNA sequencing analysis.* Counts matrix was generated from raw reads using cellranger v.2.1.0[55]. Data were analyzed using Scater[56] package for R as follows: low-quality cells were filtered out by discarding cells that failed either one of these criteria: (1) more than 10% of the cells' detected genes were mitochondrial. (2) Cells had less than 2 median absolute deviations (MADs) of detected genes (less than 455 genes). (3) Cells had less than 2 MADs log10 total counts (less than 728 counts). Next, genes that were expressed in less than ten cells were also discarded from the counts matrix. T-SNE plots were generated from the filtered counts matrix using the plotTSNE function on a subset of 99 highly variable genes with perplexity set to 20. Differential expression analysis was carried out using edgeR[57] Package for R. Genes that obtained absolute fold change greater than 2 and FDR smaller than 0.1 were considered as differentially expressed.

*B-cell Immune Repertoire sequencing.* In total, 5000–20,000 transduced CD45[+] CD10[+] and CD19[+] engrafted cells were sorted. gDNA was purified using QIAamp® DNA Micro kit (Qiagen Inc., USA). B-cell repertoire sequencing was performed using Adaptive ImmunoSEQ IGH deep assay at Adaptive Biotechnologies (Seattle, WA, USA). Analysis of B-cell receptor repertoire was done by Adaptive Biotechnologies using proprietary pipeline[58].

*Whole-exome sequencing.* Leukemic cells were collected from transplanted mice and gDNA was extracted. Whole-exome sequencing was performed using the SureSelect Human All Exon V5 + UTR kit (Agilent) as we previously described[6]. The library was paired-end sequenced on an Illumina NextSeq550 (2 × 150 bp) sequencer.

**Whole-genome sequencing.** Leukemic and BB-transduced corresponding CB cells were collected from transplanted mice and gDNA was extracted. Sequencing libraries were prepared using NEBNext ULTRA II library preparation kit (see details in Supplementary Methods) and sequenced on HiseqXten (BGI Hong Kong).

**Single-nucleotide polymorphism (SNP) array.** Array analysis was done using Affymetrix CytoScan HD array (Affymetrix, CA, USA) according to the manufacturer's recommendations (Affymetrix manual protocol Affymetrix® Cytogenetics Copy Number Assay P/N 703038 Rev. 3). The raw data were processed using Chromosome Analysis Suite (ChAS) 3.1.0.15.

**NGS sequencing of leukemic samples.** CD45[+]/CD34[+] cells from transplanted mice were sorted/magnetic beads purified. Genomic DNA and RNA were isolated (AllPrep DNA/RNA, Qiagen). cDNA was prepared from RNA. Amplicons surrounding guides for sequencing were amplified using primers with CS1-FWD and CS2-REV tales as follows (Capital letters— common sequencing tale. Lower case letters—specific sequences): Guide 1_FW_NGS: ACACTGACGACATGGTTCTACAttcgctaagtg ctcggagtt.

Guide 1_Rev_NGS: TACGGTAGCAGAGACTTGGTCTgagaatcgaagcgctacctg.
Guide 2_FW_NGS: ACACTGACGACATGGTTCTACAtagacacctggggcttgtgt.
Guide 2_Rev_NGS: TACGGTAGCAGAGACTTGGTCTgcatggttactgcctctggt.
ARFp14_NGS_Fw: ACACTGACGACATGGTTCTACAtcgtgctgatgctactgagg.
ARFp14_NGS_Rev: TACGGTAGCAGAGACTTGGTCTttctttcaatcggggatgtc.
INK4p16_NGS_Fw: ACACTGACGACATGGTTCTACAaccggaggaagaaagaggag.
INK4p16_NGS_Rev: TACGGTAGCAGAGACTTGGTCTaccagcgtgtccaggaag.

Amplified segments were subjected to a second PCR amplification in 10 μl reactions in 96-well plates to incorporate Illumina sequencing adapters and a sample-specific barcode. A mastermix for the entire plate was made using the MyTaq HS 2X mastermix. Each well received a separate primer pair with a unique 10-base barcode, obtained from the Access Array Barcode Library for Illumina (Item: 100-4876; Fluidigm, South San Francisco, CA, USA). These Access Array primers contained the CS1 and CS2 linkers at the 3′-ends of the oligonucleotides. Two μl of the reaction mixture from the first stage amplification was used as input template for the second stage reaction, without cleanup. Cycling conditions were as follows: 95 °C for 5 min, followed by ten cycles of 95 °C for 30 s, 60 °C for 30 s and 72 °C for 30 s. A final, 7-min elongation step was performed at 72 °C. Samples were pooled and sequenced on an Illumina MiSeq employing V2 chemistry (500 cycles) kit to generate 2 × 250 paired-end reads (cat. number MS-102-2003).

Data were demultiplexed using Basespace to generate 2 FASTQ files per sample. Data processing was performed with the Illumina SureSelect WTA BaseSpace application. Briefly, raw data were demultiplexed using bcl2fastq version 1.8.4. sent to NGS (Hylabs, Rechovot, Israel). Data were analyzed using CRISPResso2 online analysis tool[59].

**Western blot.** Cells (~5 × 10^6) were lysed in RIPA buffer RIPA Buffer (50 mM Tris-HCl, pH 8.0, with 150 mM sodium chloride, 1.0% Igepal CA-630 (NP-40), 0.5% sodium deoxycholate, and 0.1% sodium dodecyl sulfate) supplemented with complete protease inhibitor cocktail, kept on ice for 20 min. Lysates were then

cleared by centrifugation (12,300 × g, 10 min, 4 °C). Protein concentrations in the cleared supernatants were determined with Bio-Rad protein assay (Bio-Rad, CA) according to the manufacturer's instructions. In total, 20 μg proteins were separated by SDS–PAGE and transferred to nitrocellulose membranes. Membranes were blocked and incubated with antibodies against vinculin (1:40,000, MAB3574 millipore Sigma), CDKN2A/p16INK4A [EPR1473] (1:500, ab108349, Abcam), CDKN2A/p14ARF [EPR17878] (1:500, ab185650, Abcam), Primary antibodies were detected by fluorescent Goat anti Rabbit IgG H&L (1:10,000, IRDye® 800CW preadsorbed, ab216773, Abcam) and Goat anti-Mouse IgG H&L (1:10,000, IRDye® 800CW) preadsorbed ab216772, Abcam). Membranes were imaged using Odyssey CLx imaging system (Li-Cor, Nebraska, USA).

**Quantitative PCR.** cDNA was prepared from 100 ng RNA using SuperScript® III First-Strand Synthesis (Thermo Fisher) according to the manufacturer's protocol. Quantitative PCR was prepared using Power SYBR® Green PCR Master Mix (Thermo Fisher)with the following primer pairs: For HPRT: forward: TGA-CACTGGCAAAACAATGCA, reverse: GGTCCTTTTCACCAGCAAGCT

For p16INK4A forward: GACCCCGCCACTCTCACC, reverse: CCTGTAGGACCTTCGGTGACTGA

For p14ARF: forward: TCGTGCTGATGCTACTGAGG, reverse: GCATGGTTACTGCCTCTGGT.

Samples run and analyzed on StepOne plus thermo cycler (Thermo Fisher).

**Statistical analysis.** Data were analyzed using Microsoft Excel and GraphPad Prism software (La Jolla, CA). Data are either depicted as mean ± SE or as a scatter plot with mean ± SE. Comparisons between groups were performed by unpaired two-tailed student t-tests in two groups analysis, by one-way ANOVA tests when more than two groups were compared and groups had equal variance or in Kruskal–Wallis test—a one-way nonparametric analysis of variance, when no equal variance could be assumed. Post hoc analyses were done either by using Dunnett post hoc analysis—to compare samples to the control group or by Dunn's/Tukeys multiple comparison test to compare between all experimental groups. *P* values <0.05 were considered statistically significant.

**Reporting summary.** Further information on research design is available in the Nature Research Reporting Summary linked to this article.

## Data availability

The WGS data generated in this study have been deposited in the European Genome-phenome archive database under accession code EGAD00001005456. Data are available under restricted access for human sequences, access can be obtained by emailing t.enver@ucl.ac.uk. The WES data generated in this study have been deposited in the European Genome-phenome archive database under accession code EGAD00001007734. Data are available under restricted access for human sequences, access can be obtained by emailing ute.fischer@med.uni-duesseldorf.de. The scRNAseq data generated in this study have been deposited in the Gene Expression Omnibus under accession code GSE151126. The mass cytometry data generated in this study have been deposited in the Flow Repository under Repository ID FR-FCM-Z4XM. The bulk RNAseq data generated in this study have been deposited in the Gene Expression Omnibus under accession code GSE190070. The NGS data generated in this study have been deposited in the Gene Expression Omnibus under accession code GSE190070. The single-nucleotide polymorphism (SNP) array data generated in this study have been deposited in The European Bioinformatics Institute (EMBL-EBI) ArrayExpress—under ID E-MTAB-11258. Source data are provided with this paper.

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

## Acknowledgements

We thank Michael Gershovits and Avital Sarusi-Portuguez from the Mantoux bioinformatics institute of the Nancy and Stephen Grand Israel National Center for Personalized Medicine, Weizmann institute of science, for RNAseq analysis service, Dvir Dahary for help with WGS data analysis and Idit Shiff from the Genomic Applications Laboratory, the Core Research Facility, Faculty of Medicine—Ein Kerem, the Hebrew University of Jerusalem, Israel, for 10X RNA sequencing services. We also thank Zhaohui Gu from COH for his assistance regarding PAX5 BCP-ALL subtypes. We are indebted to Nava Gershman, Alla Zozovsky, and Itzhak Ben Moshe for help with xenograft experiments. We thank the Rawlings lab for sharing the Eu-B29 lentiviral construct. Special thanks to Liron Tuval Kochen, Dafna Gaash, Hava Rosenfeld, and Keren Shichrur from the genetic laboratory of Schneider hospital for help in leukemic sample analysis. We thank all past and present members of S.I. research group for fruitful discussions and advice. This work was supported by the Israel Science Foundation (# 1178/12 to S.I.), Children with Cancer (UK) (S.I. and T.E.), Swiss Bridge Foundation (S.I.), WLBH Foundation (S.I.), Waxman Cancer Research Foundation (S.I.), US–Israel Binational Science Foundation, Israeli health ministry ERA-NET program (#CAN-CER11-FP-127 to S.I.), Hans Neufeld Stiftung, the International Collaboration Grant from the Jacki and Bruce Barron Cancer Research Scholars' Program, a partnership of the Israel Cancer Research Fund and City of Hope (S.I. grants # 00161), the Nevzlin Genomic Center for Precision Medicine in Schneider Children's Medical Center of Israel,

The European Union's Horizon 2020 research and innovation programme under the Marie Skłodowska-Curie grant agreement No 813091 (S.I.) and the Israel Childhood Cancer Foundation (S.I.). I.G. was partially supported by Israeli ministry of Immigrant Absorption. This work was performed in partial fulfillment of the requirements for PhD degrees of Ifat Geron, Avigail Rein and Hillary Manriho, Sackler Faculty of Medicine, Tel Aviv University, Israel and of Ifat Geron in the Division of Biological Sciences University of California San Diego, USA.

## Author contributions

I.G., A.M.S., and S.I. designed the study. I.G. and A.M.S. performed most of the experiments. N.T performed initial IL7RA experiments. I.G. and H.F. performed combination IL7RA and CDKN2A CRISPR experiments. I.G. A.M.S., J.B. V.T. C.J., and A.R. performed and analyzed WGS and bulk RNAseq experiments. I.R.K. interpreted Immunoseq V(D)J rearrangements analysis. I.G., A.M.S., and O.P. performed and interpreted 10X scRNAseq. U.F., R.W., and A.B. performed and interpreted exome sequencing. J.S. and K.L.D. performed and analyzed mass cytometry experiments. Y.N.L. performed VH-region sequencing of Leukemic cells. V.M. performed Karyotype analysis of the leukemic cells M.H. performed SNP analysis. N.M. and C.W.C. provided reagents and advice for CRISPR editing experiments. I.M. provided technical support for experiments. Y.B. and C.H.M.J., and M.M. critically reviewed the experiments and provided important advice. J.P.B. and H.M. supervised and performed experiments regarding CB predisposition and provided important advice. A.N. provided cord blood samples and critically reviewed experiments. I.G., A.M.S., T.E., and S.I. analyzed and interpreted the data. I.G. and S.I. wrote the manuscript.

## Competing interests

I.R.K. is a full-time employee of Adaptive Biotechnologies, Inc. The remaining authors declare no competing interests.
