## [Peer Review File · Nature Communications]

An instructive role for IL7RA in the development of human B-cell precursor leukemiaREVIEWER COMMENTS

Reviewer #1 (Remarks to the Author); expert on B-ALL and mouse models:

Geron et al provide in their manuscript entitled „An instructive role for IL7RA in the development of human B cell precursor leukemia“ novel insight in the role of active IL7R signalling in the development of BCP-ALL. They present a human model using human cord blood transduced with IL7RA and transplanted into NSG mice. They show that IL7R signalling leads to the presence of a preleukemic cell population (CD34+CD10+CD19+) and in addition that active IL7RA pathway blocks human B-cell differentiation with accumulation of a pre-B cell population. In the transduced CD34+ CB cells JAK-STAT and mTOR signalling is active and the pre-leukemic population retains self-renewal capacity. After serial transplantation leukemia developed from one CB transduced with IL7RA. The preleukemic population was nicely characterized by scRNASeq corroborating the fingerprint of a Ph-like gene signature. Furthermore the authors nicely show that the polygenic gremlin SNPs found in this CB are highly suggestive of being supportive for BCP-ALL outgrowth.

In summary the authors demonstrate for the first time that IL7RA activation blocks B-cell differentiation in humans and solve a mystery since decades as IL7R was so far supposed to have no implication in human B-cell development. In humans IL7RA loss of function germline mutations lead to T-cell deficiency but B-cells were so far unaffected. However the presence of IL7RA activating mutations as secondary events in Ph like BCP-ALL highly suggested a role in human B-cells but prove was missing since. This manuscript elegantly closes this gap. Identification of preleucemia and tracking of these populations is from a clinical point of view of utmost importance as it allows to study conditions which allow to therapeutically control extension or regression of these populations aiming to find ways for leukemia prevention. A drawback is clearly that leukemia developed only from one single CB.

NOD SCID models always bear the problem that mice are immunodeficient and cannot reproduce entirely the clonal evolution therefore recapitulation in a transgenic murine model would be of great value but this is clearly out of the scope of the presented study and planned as a follow up by the authors.

Major

Can the authors comment on the frequent association of activating IL7RA and Pax5 loss of function mutations (either point mutations or deletions) in BCP-ALL? What is the molecular link in their view?

Did the authors perform deep sequencing in the preleukemic population (CD134+CD10+CD19+) for Pax5 P80R (or other PAX5 variants) as IL7RA mutations were described in PAX5 P80R leukemia (Gu Z et al Nat Gen 2019)?

The authors nicely show that in the CB from which BCP-ALL developed carried BCP-ALL risk SNPs such as GATA3, CEBPE or ARID5B. Can co-transfection (IL7RA and GATA/CEBPE/ARID5B) and subsequent transplantation be a way to prove the synergy in BCP-ALL development (or use of an appropriate cell line model)? Can the authors please comment on that? What was the SNP profile of the other CBs, which did not result in BCP-ALL.

Minor:

Can the authors provide an illustration or table in the supplemental information of the numbers of CBs used, in which CBs the preleukemic population was identified and from which CB finally BCP-ALL developed?

Reviewer #2 (Remarks to the Author); expert on IL7R signaling:

This is a very interesting, detailed study. The primary goal was to determine whether human cord blood HSC harboring a human IL-7R alpha chain activating mutation was sufficient to initiate the development of human B-lineage ALL in the microenvironment of NSG immune deficient mice. Although IL-7 signaling is not essential for human B cell development, sporadic reports identifying IL-7R activating mutations in a small number of human B-lineage ALL prompted the authors to determine whether a mouse human chimeric animal model could provide a new tool for examining the role of IL-7R signaling in the genesis of human B-lineage ALL.

A strength of this study is the comprehensive analysis using a wide range of methodologies. One issue that requires greater clarification is the overall methodologic/biologic(?) "efficiency" of the model. Although reference is made to various numbers of mice injected and evaluated throughout the manuscript, a clarifying statement and/or table needs to summarize the results. Specifically, how many NSG mice were injected with cord blood HSC obtained from how many donors? What % of these mice engrafted with detectable human CD19+ cells in spleen or marrow? How many of the original injected mice were used as a source of cells to initiate transfer into secondary recipients? What % of secondary recipients engrafted with CD19+ human cells and how many had evidence of genomic changes suggesting the initial stages of leukemia? How many secondary recipients were used to generate tertiary recipients? What % of tertiary recipients engrafted with CD19+ human cells and how many had evidence of genomic changes suggesting the initial stages of leukemia? And therefore and in summary, how many of the original cord blood specimens used throughout the study gave rise to human leukemia in secondary or tertiary recipients?

Minor comments-

- 1) Why were only female mice used?
- 2) Did primary mice injected with human IL-7R activating mutation transduced HSC have an increased absolute number of CD19+ cells in marrow and spleen, vis a vis similar mice injected with GFP vector transduced human HSC?
- 3) In the figures showing the various engraftment data does each symbol represent an individual mouse?
- 4) The authors need to indicate the flow cytometric phenotype of each of the subsets shown in supp Fig.2A (and elsewhere).
5. The authors should probably comment on the (likely unknown) contribution of a xenogeneic microenvironment in the development of human B-lineage ALL in their model. Might there be suppressive (or enhancing) mechanisms that reflect the impact of murine cytokines/hormones that bind to human cells that do not faithfully reflect what occurs in a human fetus or neonate?

Reviewer #3 (Remarks to the Author); expert on B-cell development:

In their manuscript, Geron et al study the role of IL7RA activating mutations in early development of human precursor-B-cell ALL (BCP-ALL). This work is based on earlier studies from this group and others who have found activating mutations in IL7RA in BCP-ALL and T-ALL. With leukemia arising from multiple hits, it remains the question what the effects of IL7RA activation is and whether this is an early hit that puts the precursor B cell population at risk for leukemia development. In this study, the authors address this issue through lentiviral transduction of IL7RA with an activating mutation in cord blood derived stem cells and subsequent transplantation into NOD-SCID-common gamma (NSG) mice. The authors show a developmental block in B-cell development in transplanted mice, which is characterized in detail with genomics and extensive mass cytometry. Moreover, the authors identify a leukemia in a secondary transplant with the mutated IL7RA construct.

The authors address an important issue and take a relevant approach with advanced techniques and systems. However, in its current form, the manuscript appears quite immature in analysis and data presentation. There is a need to show more primary data and for improved statistical analysis to

fully support the conclusions that are drawn.

Major

- The authors have performed many experiments and have put a great effort in the data. However, for the reader it is unclear how experiments were performed, how these were analyzed and how the data was put together. There is a need for schemes of experimental designs. How many times were primary transplants performed? How many CB samples were used and into how many mice were these inserted? How was data analyzed with regards to different or similar donor samples?
- Details of experimental approach and rationale: Why was the indicated PPCL insertion chosen? Where was this described and what were its effects? What was the rationale of co-expression of wt CRLF2? And what is the effect on wt expression levels of IL7RA and CRLF2? Surface expression levels of both proteins should be shown in all TD populations, as well as phosphorylation of STAT5 to illustrate activation.
- Multiple Figures, esp Figure 1 lack any primary data. How was flowcytometry analyzed? What were the gating strategies for the subsets? Furthermore, Fig 1AB and 1CD seem to present the same (but inverted) data. To me that seems redundant. It would be more interesting in further details of the subsets
- B-cell and BCP analysis. The authors base their conclusion of immaturity on IgM expression. However, B cells can also express other Ig isotypes. Are IgM- B cells really BCR negative? This needs to be confirmed by showing absence of CD79A and or CD79B, or by analysis of Ig light chain analysis.
- Similarly, interpretation of functional rearrangements should not be limited to PCR and sequence analysis of gDNA. Please confirm by staining for cytoplasmic Igmu, Iggkappa and Iglambda
- More details are needed for Ig gene rearrangements. How quantitative is the analysis in Figure 1 EF? Can the authors confirm that the missing alleles are germline and have not rearranged? Does the activating IL7RA affect the VDJ usage and/or N-nucleotide additions? These analyses should be included and discussed in the context of earlier work showing that IL7RA affects IgH and IgL rearrangements (PMID: 21680796) as well as TdT expression and N-nucleotide additions (PMID: 27658954).
- Statistics: Where shown, the data appears to be non-Gaussian distributed (Fig 1, Fig 3). Hence, the data should be represented with medians and IQR, and analyzed with non-parametric tests, i.e. Mann-Whitney, Wilcoxon rank.
- The manuscript hinges on the fact that 1 leukemia was found in 1 mouse. There is not insight into how many CBs were used and how many mice per CB were generated, and how many per condition. Hence, it is unclear if the IL7RA really has predisposed to the leukemia. Larger numbers and especially incidences of leukemia are needed to conclude this. The fact that the authors relate the leukemia to additional SNPs found in the donors quite contradicts their hypothesis that IL7RA activation instructs leukemogenesis. If that is the case, other SNPs are not needed, but the IL7RA activation leads to risk of further somatic mutations that than convert this preleukemic stage to a malignancy.

Minor:

- The introduction needs to be critically looked at. The last sentence of the first paragraph about BCR-ABL1 seems out of place. This has not been mentioned before and is not followed up on here. Furthermore, in the 3rd paragraph, the link between BCR-ABL1 and TSLP is not explained. How are these linked?
- Methods: The main methods are very succinct, some details are in the Supplement, but need to be explicitly referred to. Specifically, details of primary cell culture need to be included. References to suppl tables (esp antibody details) need to be provided. It should also be made explicit if and when we are dealing with mouse or human markers. Suggest using mCD19/hCD19 etc.
- Some Figure are quite unclear and need cleaning up, rather than copy-paste from the software output. Specifically Fig 2, Fig 3B, Fig4A, Fig 4D. Fig 4C is quite uninformative.

In conclusion, further analysis of the current data is required, as well as a better description of the experiments, results and Figures. Finally, the current n=1 leukemia is very much overinterpreted.

Response to all reviewers

We appreciate the reviewers' thoughtful comments that assisted us in improving the manuscript. Our responses to each of the three reviewers are detailed below. Major changes in the manuscript are highlighted in yellow.

In addition, for all reviewers, we wish to describe a new experiment that, although not requested by the reviewers, we believe significantly improves the manuscript.

Genomic analysis of primary human ALL with activation of IL7RA (with and without CRLF2) demonstrate that the tumor suppressor CDKN2A is almost always lost¹. This was also observed in the experimental leukemia developed in our model by transducing primary human CD34+ with activated IL7RA (IL7Rins) (Fig 3D, 4C).

We thus wished to test the hypothesis that expression of mutationally activated IL7RA and the loss of CDKN2A cooperates in transforming primary human CD34+ to B-ALL. This was confirmed by CRISPR mediated disruption of CDKN2A in human cord blood CD34+ progenitors transduced with mutated IL7RA (IL7Rins). Thus, we demonstrate, for the first time in primary human hematopoietic progenitors, that activation of IL7RA initiates B-cell preleukemia and that CDKN2A loss mediates progression to leukemia.

These experiments are described now at the last paragraph of the result section, a new Fig 5 and supplementary Fig 17-23

Changes in response to the remarks by the reviewers are highlighted in the manuscript

Detailed response to individual reviewers

Reviewer 1:

Comment 1: Can the authors comment on the frequent association of activating IL7RA and PAX5 loss of function mutations (either point mutations or deletions) in BCP-ALL? What is the molecular link in their view? Did the authors perform deep sequencing in the preleukemic population (CD34+CD10+CD19+) for PAX5 P80R (or other PAX5 variants) as IL7RA mutations were described in PAX5 P80R leukemia (Gu Z et al Nat Gen 2019)?

Response: The reviewer was interested in the status of PAX5 as previous reports demonstrated loss-of-function variants in BCP-ALL. Block of B-cell development is a hallmark of BCP-ALL and often several loss-of-function mutations in B-cell differentiation genes are concomitantly seen in the same leukemic cell (e.g. PAX5 and IKZF1 deletions). Here we show, for the first time, that IL7Rins activating mutation impair human B-cell differentiation. Presumably, it can cooperate with PAX5 loss-of-function mutation in further block of B-cell differentiation in the pathogenesis of B-ALL. Indeed, in the paper mentioned by the reviewer and cited multiple times in our manuscript, PAX5 mutations were found together with activated by IL7R and CDKN2A.

We examined the PAX5 region by exome sequencing (100X depth) and cDNA sanger sequencing of the spontaneous leukemia. The analyses showed no alteration in PAX5 (on the undeleted allele) apart from synonymous variant (C1123T) that was apparent in the germline (verified by sequencing of the BB transplanted CB cells) (see figure 1rev below). PAX5 expression is not reduced as was demonstrated by scRNAseq in the leukemic cells (see figure

2rev below) confirming that the other allele is retained. Additionally, in one of the three leukemias that were developed after transduction of IL7RA and CDKN2A disruption, we also detected deletion in PAX5.

Synonymous substitution C1123T

Figure 1Rev: synonymous variation at the remaining PAX5 allele found in Leukemia and in CB germline (GFP transduced CB from the same batch)

Figure 2Rev: Upper panel: TSNE clustering of samples after scRNAseq, lower panel: PAX5 expression shows no PAX5 expression differences between control BB transduced, Pre-Leukemic cells and Leukemic cells.

Changes in the manuscript: A paragraph with regards to PAX5 expression pattern and cDNA sequencing was added in the result chapter lines 228- and figure 2Rev added as part of supplementary figure 15B

Comment 2: Can co-transfection (IL7RA and GATA/CEBPE/ARID5B) and subsequent transplantation be a way to prove the synergy in BCP-ALL development (or use of an appropriate cell line model)? Can the authors please comment on that? What was the SNP profile of the other CBs, which did not result in BCP-ALL.

Response: We thank the reviewer for this thoughtful comment which raise a more general and fundamental question regarding the role of common germline variant in the pathogenesis of B

ALL. In the original version of the manuscript, we hypothesized that the presence of the predisposing SNPs in the specific cord blood (CB), whose transduction with IL7Rins resulted in leukemia, significantly contributed to the leukemia development. Thanks to the new experiment we performed because of the reviewer's comment, we do not think that they played a major role. We sequenced 15 batches of CBs used in our paper and discovered that these are SNPs are common (Table 1rev). Although, interestingly, only the CB in which leukemia developed had all the leukemia predisposing SNPs.

Sample	ARID5B rs10821936	CDKN2A- AS-rs564398	GATA3 rs3824662	CEBPE rs2239635	ARID5B rs7089424
1	-	+	-	-/+	-
2	-	+	-	+	-
3	-	-	-	-/+	-
4	-	-	-	-/+	-
5	-	+	-/+	-/+	-/+
6	-	-	-	-/+	-/+
*7	-/+	+	-/+	-/+	-/+
8	-/+	-	-	-/+	-/+
9	-/+	-	-	-	+
10	-/+	+	-	-	-/+
11	-	+	-	+/-	-
12	-	+	N/A	-	-
13	-/+	+	-/+	-	-/+
14	+	+	-	-/+	+
15	+	-	-	-/+	-/+
Risk allele	C	heterozygous	A	C	G

Table 1rev: SNV analysis of 15 out of 21 CB used in IL7RAins transduction experiment Batch 7 (marked with*) ((CB batch XV in new supplementary table 1) is the batch from which the leukemia developed.

As the reviewer suggested, we performed experiments to test potential synergy between IL7RA activation and the genes with the leukemia predisposing SNPs. These SNPs (especially for GATA3² which was recently specifically connected to CRLF2 Ph-like ALL (see pre-print <https://www.biorxiv.org/content/10.1101/2020.02.23.961672v1>), were described to induce a modest increased expression of these genes. We therefore designed two experiments of CEBPE/GATA3/ARID5B over expression by lentivirus vector carrying CFP. In the first, we co-transduced human CB CD34+ hematopoietic progenitors with IL7RAins and each of the genes.

We then grew the cells in B-lineage supporting culture conditions (MS5 stroma). Growth of the cells was measured after 2 weeks in culture. No significant growth advantage was observed as seen in figure 3rev.

Figure 3rev: Growth of control backbone (green) or IL7Rins (white) transduced cord blood CD34+ cells after transduction with CEBPE/GATA3/ARID5B normalized to cells transduced with empty vectors.

Additionally, we performed the same experiments on the mouse pro-B murine cell line Ba/F3. We followed the growth of the co-transduced cells. Growth was calculated per fraction of transduced (CFP positive) cells. The experiment was repeated 3 times in triplicates. As seen in figure 4rev, enhanced growth was seen in ARID 5B and GATA3 transduced cells. However, this was independent of IL7Rins as it was also seen in Ba/F3 cells transduced with backbone (BB) vector. Thus, overexpression of these genes benefits growth of the cells regardless of IL7RA activity.

Figure 4rev : Growth of BaF3 cells transduced with IL7RAins/BB and with CFP-CEBPE/GATA3/ARID5B . Graphs are representative of 3 experiments.

Important word of caution is that in these experiments the level of expression of these genes is much higher than the very modest increase of their expression caused by the SNP variant and also that it is quite possible that the predisposing variants cooperate in increasing the risks for BCP-ALL. Proper experiment would be to examine these important questions in mice (or more challengingly – human cells in mice) carrying these SNPs in the germline in various combinations. Such a fundamental and important research is beyond the scope of this current paper.

To summarize, while we cannot rule out a leukemia predisposing role for these SNPs, we believe, based on our new CRISPR editing experiments demonstrating the critical role of

CDKN2A in cooperating with IL7Rins in leukemia development, that the need for spontaneous biallelic loss of CDKN2A for leukemia development is a more plausible explanation for the rarity of leukemia development within the less than a year follow-up of the transplanted mice.

Changes in the manuscript: We removed the SNP hypothesis from the paper, as it seems too speculative. We added the CDKN2A hypothesis, which we confirmed by genomic editing, as the last paragraph of the discussion and as an illustration in figure 5C

Comment 3: Can the authors provide an illustration or table in the supplemental information of the numbers of CBs used, in which CBs the preleukemic population was identified and from which CB finally BCP-ALL developed?

Response and changes in the manuscript: We added a table and an illustration (supplementary table 1 and supplementary fig 1). Leukemia developed from CB XV. CB IX, XII and XV (marked with (P) demonstrated the CD10+CD34+ population). The same was done for the new CDKN2A in-vivo experiments (Supplementary table 6 and supplementary fig 17)

Reviewer 2:

Comment 1: overall methodologic/biologic(?) "efficiency" of the model. Although reference is made to various numbers of mice injected and evaluated throughout the manuscript, a clarifying statement and/or table needs to summarize the results.

Response and changes in the manuscript: We added a table and an illustration (supplementary table 1 and supplementary fig 1). Leukemia developed from CB batch XV. CB IX, XII and XV developed the CD10+CD34+ population). The same was done for the new CDKN2A in-vivo experiments (Supplementary table 6 and supplementary fig 17)

Comment 2: Why were only female mice used?

Response: It has been previously shown that the gender of the NOD/SCID/IL-2Rgc-null recipients has a significant impact on human hematopoietic cells engraftment with superior results in female versus male mice³ (see F. Notta, S. Doulatov, J E. Dick "Engraftment of human hematopoietic stem cells is more efficient in female NOD/SCID/IL-2Rgc-null recipients" Blood (2010) 115 (18): 3704–3707))

Comment 3: Did primary mice injected with human IL-7R activating mutation transduced HSC have an increased absolute number of CD19+ cells in marrow and spleen, vis a vis similar mice injected with GFP vector transduced human HSC?

Response: Thank you for this important comment. We evaluated the percentage CD19 of the transduced human cells rather than determining the absolute numbers of CD19+ cells in the marrows and spleen. This is due to differences in transduction efficiencies from batch to batch and different viruses. Thus, we cannot evaluate the total number of B-cell precursors at this stage, nor can we determine if the presence of IL7R activating mutation cause a relative block of B-cell differentiation or enhancement of the survival/growth of B-cell precursors. We believe that enhanced proliferation of B-cell precursors is, however, unlikely as we observed increased expression of CDKN2A (Fig 4C) and we did not observe increase expression of cell proliferation genes. We also did not detect significant oligoclonal expansion.

Changes in the manuscript: We address this issue in the highlighted sections at page 5.

Comment 4: In the figures showing the various engraftment data does each symbol represent an individual mouse?

Response: Yes. Each dot/symbol in the graphs represents a single mouse.

Changes in manuscript: Where applicable, explanation was added in the figure legend (Figure 1 and Figure 2)

Comment 5: The authors need to indicate the flow cytometric phenotype of each of the subsets shown in supp Fig.2A (and elsewhere).

Response: The gating strategy and differentiation stage analysis explaining the detailed B-cell differentiation stage is detailed in figure 5rev.

Changes in the manuscript: Previous Supplementary figure 2A is now in main manuscript as figure 1C and Figure 5rev was added as supplementary figure 5

Figure 5rev: Gating strategy of CyTOF data

Comment 6: The authors should probably comment on the (likely unknown) contribution of a xenogeneic microenvironment in the development of human B-lineage ALL in their model. Might there be suppressive (or enhancing) mechanisms that reflect the impact of murine cytokines/hormones that bind to human cells that do not faithfully reflect what occurs in a human fetus or neonate?

Response: It is generally known that the xenogeneic microenvironment in NSG mice supports human B-cell development – hence it is the most common model for xenografting human B-ALL⁴. This may be due to some cross reactivity of mIL7 with the human receptor as suggested by mouse-human cultures, in which blockage of mIL7 inhibited the growth and differentiation of human B-cells from hematopoietic progenitors⁵. In contrast the engraftment of human myeloid and T cell malignancies is better in the “NSGS” mice transgenic for several human stem cell and myeloid cytokines.

Changes in the manuscript: we added a paragraph to discuss this issue (lines 325-8, page 15)

Reviewer 3

Comment 1: There is a need for schemes of experimental designs. How many times were primary transplants performed? How many CB samples were used and into how many mice were these inserted?

Response and changes in the manuscript: We added a table and an illustration (supplementary table 1 and supplementary fig 1). The same was done for the new CDKN2A in-vivo experiments (Supplementary table 6 and supplementary fig 17)

Comment 2: Why was the indicated PPCL insertion chosen? Where was this described and what were its effects?

Response:

We originally discovered the activating mutations of IL7RA in B-ALL⁶. Our discovery has been confirmed by many other groups (Reviewed in ⁷). Most of the mutations are in-frame insertions that include cysteine into the juxtamembrane domain around a.a. 240. The presence of cysteine leads to homodimerization through S=S bonds and constitutive activation of the receptor. The insertion mutation IL7RA 243ins PPCL was one of the first we discovered, and it induces, like the other mutations, phosphorylation of STAT5 and activation of the mTOR pathway.

Comment 3: What was the rationale of co-expression of wt CRLF2?

Response:

We and others have discovered the aberrant expression of wt CRLF2 in ALL^{8,9}. CRLF2 is always co-expressed with IL7RA to form the receptor to TSLP. Moreover, we initially described the mutations in IL7RA in the context of CRLF2 aberrant expression in B-ALL⁶. Hence in the initial modeling experiment in this paper we co-expressed CRLF2 with IL7RA. Yet human CRLF2 does not interact with mouse TSLP¹⁰. Furthermore, IL7RA activating mutations are now additionally known to occur in both B and T ALLs without CRLF2 aberrations^{11,12}. Hence, we also performed experiments in which we expressed the wt and mutated IL7RA without CRLF2.

Comment 4: What is the effect on wt expression levels of IL7RA and CRLF2 Surface expression levels of both proteins should be shown in all TD populations, as well as phosphorylation of STAT5 to illustrate activation?

Response:

CRLF2 is not normally expressed in B-cell progenitors hence there is no endogenous expression besides the expression following the transduction. As for IL7RA, membranal expression of IL7RA in transduced CB according to flow cytometry is overall low. This actually confirms that our experiment did not result in dramatic non-physiologic expression of IL7RA. See in gating strategy (figure 6Rev untransduced CB showing endogenous IL7RA expression vs IL7RA-GFP transduced CB

Comment 5: How was flowcytometry analyzed? What were the gating strategies for the subsets?

Response and changes in the manuscript: We added a gating strategy supplementary figure 3

Figure 6rev: gating strategy for Immunophenotyping analysis

Comment 6:

Fig 1AB and 1CD seem to present the same (but inverted) data. To me that seems redundant. It would be more interested in further details of the subsets

Response and change in the manuscript: We accepted the reviewer's comment The figure now includes only non-redundant data with details of subsets by mass cytometer. For subsets see our response to the following comment

Comment 7: B-cell and BCP analysis. The authors base their conclusion of immaturity on IgM expression. However, B-cells can also express other Ig isotypes. Are IgM-B-cells really BCR negative? This needs to be confirmed by showing absence of CD79A and or CD79B, or by analysis of Ig light chain analysis. Similarly, interpretation of functional rearrangements should not be limited to PCR and sequence analysis of gDNA. Please confirm by staining for cytoplasmic Igm μ , Igkappa and Iglambda

Response:

We obviously agree with the reviewer. Relative immaturity of the transduced population was initially based on IgM- expression on CD19+ cells. We then further expanded the analysis to reflect CD10+IgM- of the transduced CD19+ cells which better reflect B-Cell progenitors. Additionally, the immaturity of the cells was further verified by mass cytometry analysis (as previously described by Good et al¹³). New Fig 1C depicts the actual B-cell differentiation stage of the transduced cells. The mass cytometry data agrees with the flow cytometry IgM analysis.

Furthermore, we stained the four BCP-ALL leukemias for expression IGK/IGL (Figure 7rev).

Figure 7Rev: Immunophenotyping of 4 normal CB and 4 Leukemias from different CB batches.

Changes in the manuscript: The mass-cytometry (CyTOF) analysis is now depicted in Fig 1C. Supplementary figure 5 was added to explain gating strategy for CyTOF analysis.

Previous fig 1 B,D were corrected to reflect only CD10+ IgM- fraction and are now in supplementary figure 4A,B

Comment 8:

More details are needed for Ig gene rearrangements. How quantitative is the analysis in Figure 1 EF? Can the authors confirm that the missing alleles are germline and have not rearranged? Does the activating IL7RA affect the VDJ usage and/or N-nucleotide additions? These analyses should be included and discussed in the context of earlier work showing that IL7RA affects IgH and IgL rearrangements (PMID: 21680796) as well as TdT expression and N-nucleotide additions (PMID: 27658954).

Response:

1. With regard to the quantitative nature of the assay in Figure 1 EF : The analysis was based on sequencing of VDJ rearrangement of sorted CD19⁺ and CD10⁺ cells from three CRLF2-IL7RA transduced and three BB transduced samples. The sequencing was performed using Adaptive ImmunoSEQ IGH deep assay at Adaptive Biotechnologies. This assay is the most quantitative assessment of sequence-based immune receptor profiling that currently exists. Attached please find links to two papers (Carlson et. al. <https://www.nature.com/articles/ncomms3680> and Ching et. al. <https://bmccancer.biomedcentral.com/articles/10.1186/s12885-020-07077-9>) which provide information about the analytic validity of the B (Ching et al) and T-cell assays.

The Ching et. al. paper is a compilation of the work that supported the FDA de novo 510 K clearance of the assay in 2018. Any assay developed by Adaptive Biotech includes the creation of a synthetic immune system that represents every possible V-J rearrangement (based on IMGT) for whatever locus is being assessed. This synthetic immune system allows iterative primer balancing to control for PCR bias. These same templates are added into the individual reactions to provide internal controls for PCR amplification.

2. Regarding the issue of germline status of certain alleles. (Assuming that the reviewer aims for the VDJ analysis of the 6 samples and not the Leukemia analysis) - Again as in #1, every V-J rearrangement is interrogated. The presence of a fraction of ALL clones that are too developmentally immature to have undergone V(D)J rearrangement prior to transformation has been well documented. It was initially appreciated by Southern blot analysis in which all or some of the immune receptor loci that were analyzed appear to be retained in their germline configuration as described in^{14,15 16}.
3. Regarding the effect of activated IL7RA on N-nucleotide addition and VDJ usage. We thank the reviewer for the referral to the issue. A detailed analysis of the sequenced cells demonstrated enhanced insertions of N-nucleotides in the IL7RAins transduced cells compared to control transduced cells (see figure 8rev). The increased N nucleotide addition is consistent with the increased expression of TdT (DNMT) in the preleukemic population. See fig 9rev also added in supplementary figure 15.

Figure 8rev: Histograms represent average % of reads for each N- nucleotides insertion quantity was measured of each condition (Backbone /CRLF2-IL7RAins transduced transplanted CD19/CD10 sorted cells n=3 for each). Difference was found significant $p < 0.0001$ in repeated measurements ANOVA test.

Figure 9 rev Expression of DNTT in Leukemia, pre-Leukemia and BB transduced control as analyzed after single cell 10X RNAsequencing

Additionally, we identified preferential usage of proximal IGHV genes as was previously reported in fetal B-cells. The latter was additionally supported by analysis of bulk RNA sequencing of five paired CRLF2-IL7RAins vs BB control.

Changes in the manuscript:

The N nucleotide data was implemented in the result section page 5 lines 114-118 and in Supplementary figure 4C. DNTT expression was added in supplementary figure 15 and added to the discussion (lines 341-3)

Comment 9 :Statistics: Where shown, the data appears to be non-Gaussian distributed (Fig 1, Fig 3). Hence, the data should be represented with medians and IQR, and analyzed with non-parametric tests, i.e. Mann-Whitney, Wilcoxon rank.

Response: Indeed, in some cases the data was non-Gaussian distributed. Mann-Whitney is used for two group comparisons and not multiple groups. When comparing multiple groups, Kruskal-Wallis test - a one-way non-parametric analysis of variance, can be used – as was done in this paper for multiple groups when no equal variance could be assumed.

Comment 10: The manuscript hinges on the fact that 1 leukemia was found in 1 mouse. There is not insight into how many CBs were used and how many mice per CB were generated, and how many per condition. Hence, it is unclear if the IL7RA really has predisposed to the leukemia. Larger numbers and especially incidences of leukemia are needed to conclude this.

Response:

We absolutely agree with the reviewer and hence performed the additional experiments described in our response to all the reviewers in the beginning of this document. Indeed, primary leukemia developed in one out of eight CB batches that were transduced with IL7RAins. In the revised manuscript we show evidence of 3 more leukemias that were developed more rapidly from 3 out of seven CB batches that were co-transduced with activated IL7RAins with a CRISPR vector designed to eliminate CDKN2A, as is often seen in patients with ALL and IL7RA activation. Together these results demonstrate, for the first time in primary human hematopoietic cells, that activation of IL7RA can initiate a preleukemic condition that cooperates with the loss of CDKN2A to induce B-cell precursor ALL. Numbers of the CB that were used and mice that were transplanted are now summarized both in Supplementary figures 1,17 and Supplementary tables 1,6.

Changes in Manuscript:

A new section describing additional *in vivo* CRISPR experiment with new leukemias was added at the end on the the result section with supporting figures. This issue is also detailed in the last paragraph of the discussion.

Comment 11: The fact that the authors relate the leukemia to additional SNPs found in the donors quite contradicts their hypothesis that IL7RA activation instructs leukemogenesis. If that is the case, other SNPs are not needed, but the IL7RA activation leads to risk of further somatic mutations that than convert this preleukemic stage to a malignancy.

Response: While we decided to omit this section completely (see detailed response to reviewer 1), we feel that we owe the reviewer explanation that relates to the current general theory of the pathogenesis of childhood lymphoblastic leukemia. Leukemia is initiated by a major “initiating” oncogenic event that occurs in a single cell. The progeny of this cell constitutes the “preleukemic clone”. For B-ALL examples include the ETV6-RUNX1 translocation or as we show in the current manuscript, by activating mutation in IL7RA. Progression to leukemia usually requires additional events – for example deletion of CDKN2A, PAX5, IKZF1, ETV6, EBF1 or oncogenic mutations in JAK2, RAS etc. In our manuscript we demonstrate the critical role of CDKN2A expression.

GWAS studies also revealed that there are relatively common variants (SNPs) that modestly increase the risk to leukemia (e.g. X1.5). Still most people carrying these SNPs will never develop leukemia. For example, it was discovered that a common SNP in the region of GATA3, that leads to modest increased expression of GATA3, enhances the risk for CRLF2/IL7R ALL. New pre-print paper suggests that GATA3 variants alter chromatin topology thus increasing the risk for B-ALL, and specifically Ph-like B-ALL (H. Yang et al.

<https://www.biorxiv.org/content/10.1101/2020.02.23.961672v1.full.pdf>).

To summarize – one has to distinguish between *predisposing germline genomic variants* to *initiating somatic genomic events*. The GATA3 SNP is predisposing and the IL7R mutation is initiating.

As leukemia developed only in one CB transduced with IL7RAins we hypothesized that this may relate to predisposing SNPs. Reviewer 1 smartly asked us to test the SNPs in the other CB that were not transformed to leukemia. As we found that the SNPs were common also in CB that

were not transformed, we have abandoned this hypothesis. As detailed in the last paragraph of the discussion, we believe that the singularity of the initial leukemia was due to the rare occurrence of biallelic loss of CDKN2A and, possibly, the acquiring of IKZF1 mutations.

Changes in the manuscript: We removed the SNP hypothesis from the paper.

Comment 12: introduction needs to be critically looked at. The last sentence of the first paragraph about BCR-ABL1 seems out of place. This has not been mentioned before and is not followed up on here. Furthermore, in the 3rd paragraph, the link between BCR-ABL1 and TSLP is not explained. How are these linked?

Response:

BCR-ABL1 (coded by the t(9;22) translocation that is also termed “Philadelphia chromosome”) driven BCP-ALL is a subtype of BCP-ALL that is characterized by unique kinase activated expression pattern. Deregulation of the TSLP receptor components (CRLF2 aberrant expression and IL7RA mutations) are found in a subtype of BCP-ALL with similar expression pattern to Philadelphia BCP-ALL and is thus termed “Ph-Like” or “BCR-ABL like” BCP-ALL. This manuscript discusses the relevance of IL7RA to the development of Ph-Like BCP-ALL.

Changes in the manuscript: We rephrased the relevant paragraphs to better explain the connection between BCR-ABL1 and Ph-Like BCP-ALL lines 72-75.

Finally, we addressed all the reviewer comments regarding editing and method details. Once the manuscript is accepted to Nature Communications formatting of the paper and the figures will be adjusted to the journal requirements.

Many thanks for this reviewer exquisite detailed review.

Cited references

- 1 Gu, Z. *et al.* PAX5-driven subtypes of B-progenitor acute lymphoblastic leukemia. *Nature Genetics*, doi:10.1038/s41588-018-0315-5 (2019).
- 2 Perez-Andreu, V. *et al.* Inherited GATA3 variants are associated with Ph-like childhood acute lymphoblastic leukemia and risk of relapse. *Nat Genet* **45**, 1494-1498, doi:ng.2803 [pii] 10.1038/ng.2803 (2013).
- 3 Notta, F., Doulatov, S. & Dick, J. E. Engraftment of human hematopoietic stem cells is more efficient in female NOD/SCID/IL-2Rgc-null recipients. *Blood* **115**, 3704-3707, doi:10.1182/blood-2009-10-249326 (2010).
- 4 Townsend, E. C. *et al.* The Public Repository of Xenografts Enables Discovery and Randomized Phase II-like Trials in Mice. *Cancer Cell* **29**, 574-586, doi:10.1016/j.ccell.2016.03.008 (2016).
- 5 Parrish, Y. K. *et al.* IL-7 Dependence in human B lymphopoiesis increases during progression of ontogeny from cord blood to bone marrow. *J Immunol* **182**, 4255-4266, doi:182/7/4255 [pii]

10.4049/jimmunol.0800489 (2009).

6 Shochat, C. *et al.* Gain-of-function mutations in interleukin-7 receptor- α (IL7R) in childhood acute lymphoblastic leukemias. *J Exp Med* **208**, 901-908, doi:jem.20110580 [pii]

10.1084/jem.20110580 (2011).

7 Tal, N., Shochat, C., Geron, I., Bercovich, D. & Izraeli, S. Interleukin 7 and thymic stromal lymphopoietin: from immunity to leukemia. *Cell Mol Life Sci* **71**, 365-378, doi:10.1007/s00018-013-1337-x (2014).

8 Cario, G. *et al.* Presence of the P2RY8-CRLF2 rearrangement is associated with a poor prognosis in non-high-risk precursor B-cell acute lymphoblastic leukemia in children treated according to the ALL-BFM 2000 protocol. *Blood* **115**, 5393-5397, doi:blood-2009-11-256131 [pii]

10.1182/blood-2009-11-256131 (2010).

9 Hertzberg, L. *et al.* Down syndrome acute lymphoblastic leukemia, a highly heterogeneous disease in which aberrant expression of CRLF2 is associated with mutated JAK2: a report from the International BFM Study Group. *Blood* **115**, 1006-1017, doi:blood-2009-08-235408 [pii]

10.1182/blood-2009-08-235408 (2010).

10 Francis, O. L. *et al.* A novel xenograft model to study the role of TSLP-induced CRLF2 signals in normal and malignant human B lymphopoiesis. *Haematologica*, doi:10.3324/haematol.2015.125336 (2015).

11 Reshmi, S. C. *et al.* Targetable kinase gene fusions in high-risk B-ALL: a study from the Children's Oncology Group. *Blood* **129**, 3352-3361, doi:10.1182/blood-2016-12-758979 (2017).

12 Zenatti, P. P. *et al.* Oncogenic IL7R gain-of-function mutations in childhood T-cell acute lymphoblastic leukemia. *Nat Genet* **43**, 932-939, doi:ng.924 [pii]

10.1038/ng.924 (2011).

13 Good, Z. *et al.* Single-cell developmental classification of B cell precursor acute lymphoblastic leukemia at diagnosis reveals predictors of relapse. *Nature medicine* **24**, 474-483, doi:10.1038/nm.4505 (2018).

14 Felix, C. A. *et al.* Characterization of immunoglobulin and T-cell receptor gene patterns in B-cell precursor acute lymphoblastic leukemia of childhood. *J Clin Oncol* **8**, 431-442, doi:10.1200/JCO.1990.8.3.431 (1990).

15 Felix, C. A. *et al.* Immunoglobulin and T cell receptor gene configuration in acute lymphoblastic leukemia of infancy. *Blood* **70**, 536-541 (1987).

16 van der Burg, M. *et al.* Immunoglobulin light chain gene rearrangements display hierarchy in absence of selection for functionality in precursor-B-ALL. *Leukemia* **16**, 1448-1453, doi:10.1038/sj.leu.2402548 (2002).

REVIEWER COMMENTS

Reviewer #1 (Remarks to the Author):

The authors have extensively extended the experimental workup and the scope of their article. Especially the addition of experimental data on the role of CDKN2A loss and progression of IL7RA pB-ALL is very interesting, experimentally deeply analyzed and extends the understanding of initiation and progression of pB-ALL.

I fully agree to the modifications the authors suggest and appreciate removal of the SNP hypothesis in benefit of the role of CDKN2A loss in IL7Rins pB-ALL.

All my comments have been addressed and in light of the novelty and extension of results I consider this manuscript of high importance for the readership of Nature communications.

Reviewer #2 (Remarks to the Author):

The authors have gone to great lengths to improve their study and provided strong and satisfactory comments to my concerns. Particularly important is the inclusion of new data indicating a cooperative relationship between IL-7R activating mutations and loss of CDKN2A in the development of BCP-ALL.

Very well done!

Reviewer #3 (Remarks to the Author):

I thank the authors for carefully addressing the issues that I had raised.

Reviewer #4 (Remarks to the Author):

This paper presents exciting original data demonstrating a role for activation of signalling in the initiation of BCP-ALL from primary human haematopoietic cells. The revisions made to date have added additional mechanistic detail and improved the clarity of reporting. The work presented will be a valuable addition to the leukaemia pathogenesis field. I would recommend the following points are considered prior to publication.

Major points

1. Regarding the CD10hi population

-The flow cytometry data, as presented, does not convince that a CD10hi population is found in more than one sample. It is reported that 6 animals had such a population. It is not possible to see this in Supplementary Figure 6 as the CD10hi gate appears pasted over the data and is obscuring it. In Fig 2c, the CD10hi populations look different in CRLF2-IL7RAins and IL7RAins, both in terms of SSC and CD10 expression. There is insufficient data in the CRLF2-IL7RAins plots to convince that the proportion of CD34 expressing cells is greater. IL7RAins is convincing (though difficult to see, can you make the data points larger?). We are referred to Supplementary table 1 but there is no info about the CD10hi population here.

-How many mice had a CD10hi population with the same immunophenotype as in the bottom panel fig 2B (SSC, CD10, CD34 intensity).

-At the moment, there is not sufficient data to support the claim (e.g. line 144) that the CD10hi

population has a pre-leukaemic phenotype. The classifier is used to show that the leukaemia is pro-BII like. Should the same method not be applied to the pre-leukaemia? In both cases, there is a missed opportunity to show which epitopes differentiate the pre-/leukaemia differ from a typical pro-BII cell.

-A more precise use of terminology would assist the reader. The work 'pre-leukaemia' is used to refer to the state in the mouse, the CD10+CD19+ population and the CD10hi population at present.

2. Analysis of RNAseq data

-It is very confusing to show a 3-way Venn diagram of DEGs when 2 of the samples overlap. Can you remove the CD10hi fraction (for example by using an MME expression cutoff) from the CD10+CD19+ pre-leukaemia and redo?

-The numbers of DEGs are helpful to build up a picture of hierarchical relationships but there needs to be some elaboration of what these genes are e.g. GO annotations, to help elucidate what changes take place en route to pre-leukaemia and leukaemia.

-Have the authors attempted any trajectory analysis to add additional support to their claims about hierarchy?

-The GSEA presented in Fig 4d is an indirect way of making a comparison to Ph-like B-ALL. More direct methods would include providing AGA distances in combined Euclidian space, showing a module score of marker genes (like the ones in 4e), or regression e.g. logistic regression, with the latter being the strongest evidence as it uses the entire transcriptome profile.

-The TSNE map is not well-suited to comparing marker gene expression between clusters because the clusters are overlapping. It would be better to show a violin plot or dotplot.

-The expression patterns in 4e lack context. We need to know what the relative expression in Ph-like BCP-ALL is.

-Is the mouse that gave rise to the leukaemia included in the bulk RNAseq? How many of these samples have a CD10high population?

Minor points

Fig 1- Why are these specific statistical comparisons shown? Are those not shown not significant or not tested? Data points should be shown for panels c, d, e.

Fig 2- Labels or legends for panel b and c are switched. Why does number of data points in b not match 'n' in legends? Missing y-axis label for CD34 plots. Please state in fig/ legend what gates precede the plots shown in the hierarchy e.g. live, single cells.

Fig 3- Panel a conveys no morphological information- too small and magnification too low. In panel b, it would be helpful to see the pre-leukaemia plotted alongside with these axes.

Fig 4- Panel d. How can the x-axis be both ILR7RA-BB and Ph-like non-Ph like?

Supp Fig 1- useful to include what cells were harvested for secondary transplant and how many

Supp Fig 2- how many times performed?

Supp Fig 5- the legend says PBMC are shown, but the methods state it is bone marrow (which it must be to have these B progenitor populations in).

Supp Fig 6- the CD10hi gate is obscuring the data.

Supp Fig 7- why is untransplanted mouse shown? Wouldn't pre-leukaemia or BB control be more relevant if the point is that spleen/BM are infiltrated with human CD45+ in leukaemia?

Supp Fig 14- is mislabelled Sup Fig 1. How many times performed? Why does BB have a stronger response to Il-7 than leukaemia? Is that expected?

Supp Figure 15- is mislabelled Sup Fig 2. It should probably show RAG2 as well

We have been delighted to read that this new reviewer joined the previous three reviewers in their excitement from our paper, especially their uniformed conclusion the “exciting original data” presented in our paper “will be a valuable addition to the leukaemia pathogenesis field”. We also appreciate the constructive new comments by this reviewer that assisted us in further improving the manuscript. Herein is a point-by-point response to these comments and detailing the resulted improvement in the manuscript.

Comment 1: Regarding the CD10hi population

a)The flow cytometry data, as presented, does not convince that a CD10hi population is found in more than one sample. It is reported that 6 animals had such a population. It is not possible to see this in Supplementary Figure 6 as the CD10hi gate appears pasted over the data and is obscuring it. In Fig 2c, the CD10hi populations look different in CRLF2-IL7RAins and IL7RAins, both in terms of SSC and CD10 expression. There is insufficient data in the CRLF2-IL7RAins plots to convince that the proportion of CD34 expressing cells is greater. IL7RAins is convincing. We are referred to Supplementary table 1 but there is no info about the CD10hi population here.

Response:

Regarding supplementary figure 6, it was unfortunately corrupted probably after conversion from word document to PDF. This was now fixed and the CD10high population is visible. (See below)

As for the apparently different appearance of the CD10hi populations in CRLF2-IL7RAins and IL7RAins, we did see several patterns of CD10 high populations independent from the presence of CRLF2. All can be now seen clearly in supplementary figure 6. Three of the samples had pattern that more resembled the mouse that originated the leukemic cells (that were IL7Rins), and were originated from CRLF2-IL7RAins. However, the CD10 intensity, was highest in the pre-leukemic mouse.

Changes in the manuscript: The figure was re-drawn with a compatible program and now the CD10high population can be clearly seen. Same was done for figure 2C.

Supplementary figure 6

Figure 2

Regarding the proportion of CD34 expressing cells, it was not greater in the total CRLF2-IL7RAins engrafted cells, nor was this claimed in the paper (see graph in figure 2a, showing that only IL7RAins alone had overall more CD34CD10 cells). However, some of the engrafted cells (from batches IX, XII, XVI - marked with (P) in supplementary table 1) had high CD10 population. When gating on the CD10high population of samples that had this population, the CD34% was higher than in the total CD10⁺CD19⁺ population (See fig 1rev in which we show engraftment of transduced cells from 3 batches). As we do not have CD34 staining for all these populations (some are old samples with no available material for re-stain), we did not make any statement as for the CD34 expression in the CRLF2-IL7RAins high CD10 population.

Figure rev1:

Flow cytometry plot of 3 CB batches with CD10^{high} population that were stained with CD34 (APC). Left panel presents the CD10 CD19 distribution (gated on live (7AAD) human (hCD45 vio-green) transduced [CRLF2 (PE) for CRLF2-IL7RA cells /IL7RA (Brilliant violet 421) for IL7RA cells / GFP for BB] cells. Mid panel presents CD34 staining for total CD10⁺CD19⁺ population. Right panel presents CD34 staining for CD10^{high} population when present. (a) Top: batch XII, bottom: batch IX. (b) Batch XVI (cage 2) (C) Batch XVI (cage 3)

b) At the moment, there is not sufficient data to support the claim (e.g. line 144) that the CD10^{hi} population has a pre-leukaemic phenotype. The classifier is used to show that the leukaemia is pro-BII like. Should the same method not be applied to the pre-leukaemia? In both cases, there is a missed opportunity to show which epitopes differentiate the pre-/leukaemia differ from a typical pro-BII cell.

-A more precise use of terminology would assist the reader. The work 'pre-leukaemia' is used to refer to the state in the mouse, the CD10⁺CD19⁺ population and the CD10^{hi} population at present.

We agree with the reviewer that the term "pre-leukemia" is used imprecisely with different

meanings. Furthermore, as the rate of spontaneous progression to leukemia was very low, the actual potential of specific leukemogenic potential of “pre-leukemic” populations could not be assessed. As detailed below we have now carefully modified the general use of this term in the manuscript.

As for our suggestion that the CD10^{high} population has a pre-leukemic phenotype: the classifier was used to elucidate the effect of transduction on transduced cells’ differentiation. It was not performed on many samples and on the original sample from the pre-leukemic mouse due to shortage of cells. Nevertheless, based on the VDJ sequencing in which the CD10^{high} population was enriched with the pre-leukemic clone combined with the distinct expression pattern that had higher resemblance to the leukemia than the CD10⁺ CD19⁺ population, we believe that the preleukemic population that resulted in the development of the spontaneous leukemia, originated in the CD10^{high} fraction. Furthermore, the fact that CD10^{high} population in IL7RAⁱⁿ transduced cells and was enriched with CD34 expressing cells strengthen the notion that this population may represent a “pre-leukemic” population. However we agree with the reviewer that there is not sufficient data to fully generalize this claim. Thus, in the revised paper we reserved the term “preleukemia” mainly to the population derived from the preleukemic mouse (the mouse whose cells evolved to leukemia upon secondary transplantation).

Changes in the manuscript:

We have modified the use of the term “pre-leukemia” throughout the manuscript.

Examples:

Title of section in line 127 was changed from: “Aberrant expression of activated IL7RA induces a pre-leukemic B-cell precursor immunophenotype retaining self-renewal capacity” to “Aberrant expression of activated IL7RA induces a B-cell precursor population that retains self-renewal capacity.”

The sentence in line 144 that does not refer to CD10 expression but rather to the differentiation arrest and self-renewal properties, was modified to precisely reflect these points.

Title in line 196 was changed from “Single cell analysis of B-cell precursor cells transduced with activated-IL7RA reveals a pre-leukemic population with a strong Ph-like pro-survival gene signature.” To: “Single cell analysis of B-cell precursor cells transduced with activated-IL7RA reveals a distinct population with a strong Ph-like pro-survival gene signature.”

Comment 2. Analysis of RNAseq data

a) It is very confusing to show a 3-way Venn diagram of DEGs when 2 of the samples overlap. Can you remove the CD10^{hi} fraction (for example by using an MME expression cutoff) from the CD10⁺CD19⁺ pre-leukaemia and redo?

Response:

Thank you for this very important comment! Indeed, we overlooked it. Venn diagram was recalculated after using high MME and CD34 expression to differ between the CD10⁺CD19⁺ and CD10^{high} populations.

Changes in the manuscript:

Numbers in the text were changed. Venn diagram was updated and added to figure 4 (as Figure 4C).

Figure 4C

b) The numbers of DEGs are helpful to build up a picture of hierarchical relationships but there needs to be some elaboration of what these genes are e.g. GO annotations, to help elucidate what changes take place en route to pre-leukaemia and leukaemia.

Response:

We thank the reviewer for pointing out this option that helped focusing the nature of the CD10high population and it's pre-leukemic potential. We have formed a list of the DEGs (added as excel file to supplementary data -see attached file "DEGs in Venn" for reviewer) and used it to perform a functional annotation using DAVID Bioinformatics Resources 6.8 (David.ncifcrf.gov). Main conclusion from this analysis was added to the result section.

Changes in the manuscript:

A file with the list of DEGs will be added to the supplementary material. A paragraph describing functional analysis was added in the text. And a figure with main results from the analysis was added as Supplementary figure 15.

Supplementary figure 15

c) Have the authors attempted any trajectory analysis to add additional support to their claims about hierarchy?

Response:

We attempted trajectory analysis (Monocle3), however this was not too revealing as the leukemic cells were extended throughout the time-line. (see Figure rev2). As 10x genomics data included about 10,000 cells from 4 different mice with ~ 1000 genes per cell, technical difficulties with such analysis were expected by our bioinformatician who does not feel that this analysis biologically represents our data.

Figure rev2: trajectory analysis using Monocle3 was attempted with Leukemia sample and samples from pre-leukemic mouse (no BB control). Pseudo-time 0 is at the Leukemia sample, at the middle of the graph (mark red).

d) The GSEA presented in Fig 4d is an indirect way of making a comparison to Ph-like B-ALL. More direct methods would include providing AGA distances in combined Euclidian space, showing a module score of marker genes (like the ones in 4e), or regression e.g. logistic regression, with the latter being the strongest evidence as it uses the entire transcriptome profile.

Response:

As the reviewer suggested, AUCell (Area Under the Curve for single cell) analysis was performed. This analysis calculates whether a critical subset of the input gene set is enriched within the expressed genes for each cell. Each cell is assigned a score that portrays k how much it expresses the genes of a given set (reference: Aibar S, Bravo Gonzalez-Blas C, Moerman T, Huynh-Thu V, Imrichova H, Hulselmans G, Rambow F, Marine J, Geurts P, Aerts J, van den Oord J, Kalender Atak Z, Wouters J, Aerts S (2017). "SCENIC: Single-Cell Regulatory Network Inference And Clustering." *Nature Methods*, **14**, 1083-1086. doi: [10.1038/nmeth.4463](https://doi.org/10.1038/nmeth.4463).)

Changes in the manuscript:

Figure 4 was changed to include the above analysis and paragraph was added to describe the results.

e) The TSNE map is not well-suited to comparing marker gene expression between clusters because the clusters are overlapping. It would be better to show a violin plot or dotplot. -The expression patterns in 4e lack context. We need to know what the relative expression in Ph-like BCP-ALL is.

Response:

As the reviewer suggested, a Violin plot was prepared for each of the genes in figure 4e.

Regarding the context of the expression in 4e and the relative expression in Ph-like BCP-ALL; We based our observation on a clinically validated LDA diagnostic test that identified a list of 15 genes with relative elevated expression specifically in Ph and Ph-like compared to non Ph and Ph-like ALL (see Harvey R. C et al. "Development and Validation Of a Highly Sensitive and Specific Gene Expression Classifier To Prospectively Screen and Identify B-Precursor Acute Lymphoblastic Leukemia (ALL) Patients With a Philadelphia Chromosome-Like ("*Ph-like*" or "*BCR-ABL1-Like*") Signature For Therapeutic Targeting and Clinical Intervention" <https://doi.org/10.1182/blood.V122.21.826.826>. Harvey R. C, S.K Tasian "Clinical diagnostics and treatment strategies for Philadelphia chromosome-like acute lymphoblastic leukemia" <https://doi.org/10.1182/bloodadvances.2019000163>)

In the single cell analysis we saw that the relative expression of these genes in the leukemia/CD10high/IL7RAins transduced group is elevated compared to the BB control group. Since we cannot estimate the relative expression of our samples to those in Ph-like cells, but do think that this finding supports the clinical relevance of our model we moved it to supplementary material section.

Changes in the manuscript:

The expression analysis was moved to supplementary figure 17. Violin plot was added as Supplementary figure 18

f) Is the mouse that gave rise to the leukaemia included in the bulk RNAseq? How many of these samples have a CD10high population?

Response:

The bulk RNAseq was done on CRLF2-IL7RAins transplanted mice and the mouse which gave rise to the leukemia was not in this group. 3 of the samples had CD10high population.

Minor points

Fig 1- Why are these specific statistical comparisons shown? Are those not shown not significant or not tested?

Response

Correct, those that are not shown have no statistically significant.

Data points should be shown for panels c, d, e.

Response:

For panels d,e the figure has been changed and now is presented as dot plot. For panel c, we did the same (see attached dot plot figure rev 3). However, we think that the bar graphs is easier to view by the reader and better represents the data, thus the graph in the figure was not changed.

Figure rev 3

Changes in the manuscript:

Panels d and e in Figure 1 were changed to dot plot.

Figure 1

Fig 2- Labels or legends for panel b and c are switched. Why does number of data points in b not match 'n' in legends? Missing y-axis label for CD34 plots. Please state in fig/ legend what gates precede the plots shown in the hierarchy e.g. live, single cells.

Response:

As for panels labeling, an error was made and it was fixed.

As for the number of samples on plot, it differed from the stated n, since 6 samples had a zero value and were out of the range in logarithmic graph. (in a previous version of the manuscript we presented a non- logarithmic graph with breaks on the y axis, however a reviewer suggested that a logarithmic presentation is more suitable).

Changes in the manuscript:

Figure 2 was re-made with correct labeling (see in response to comment 1).

An explanation for the discrepancy between the numbers of the dots visible on the plot and the stated n was added in the figure legend.

Fig 3- Panel a conveys no morphological information- too small and magnification too low. In

panel b, it would be helpful to see the pre-leukaemia plotted alongside with these axes.

Response:

As for panel a, since no fresh material for re-staining is available (only frozen material) the figure was moved to supplementary material.

Panel b pre-leukemia staining was added.

Changes in the manuscript:

Previous panel a of figure 3 was added to supplementary figure 7. Staining of cells from the Pre-leukemic mice was added.

Fig 4- Panel d. How can the x-axis be both ILR7RA-BB and Ph-like non-Ph like?

Response:

We thank the reviewer for noticing this mistake. The ILR7RA-BB refers to the curve distribution while the Ph-like non-Ph like refers to the gene list and should be in the title of the analysis. This was now changed.

Changes in the manuscript:

Panel D of figure 4 is now panel E and the labeling was corrected.

Supp Fig 1- useful to include what cells were harvested for secondary transplant and how many

Changes in the manuscript:

The information was added in the figure legend.

Supp Fig 2- how many times performed?

Response

This pSTAT5 analysis was performed to validate the activity of the transduced receptor. This was previously shown (with a different vector) in previous publications from our group (e.g Shochat, C. *et al.* Gain-of-function mutations in interleukin-7 receptor- $\{\alpha\}$ (IL7R) in childhood acute lymphoblastic leukemias. *J Exp Med* **208**, 901-8 (2011), Hertzberg, L. *et al.* Down syndrome acute lymphoblastic leukemia, a highly heterogeneous disease in which aberrant expression of CRLF2 is associated with mutated JAK2: a report from the International BFM Study Group. *Blood* **115**, 1006-17 (2010)). To validate the activity of the altered vector, flow analysis was done once in O18Z cells, and then repeated in BaF3 cells. (see figure rev 4) additionally we analyzed downstream activity of the receptor in different combinations (CRLF2 and IL7RA) using western blots, thus it is clear that the receptor encoded by the virus is active.(differences in activity between the cell lines is expected due to inter-species protein interactions)

Figure rev 4: transduced and sorted BaF3 cells were starved from cytokines and serum for 5 hours, followed by 20 minutes of TSLP or IL7 (4ng/ml) stimulation or no stimulation. Cells were then immediately fixed with PFA followed by methanol fixation and stained for pSTAT5.

Supp Fig 5- the legend says PBMC are shown, but the methods state it is bone marrow (which it must be to have these B progenitor populations in).

Response

Thank you for noticing the mistake. There was a mistake in the legend that was fixed.

Changes in the manuscript

The legend was fixed and now it states that BM was used.

Supp Fig 6- the CD10hi gate is obscuring the data.

Changes in the manuscript: The figure was recreated with a compatible program and now the CD10high population can be clearly seen (see in response to comment 1)

Supp Fig 7- why is untransplanted mouse shown? Wouldn't pre-leukaemia or BB control be more relevant if the point is that spleen/BM are infiltrated with human CD45+ in leukaemia?

Response

Untransplanted mouse is the negative control (By which the CD45 gate is drawn) we did not have tertiary transplants from BB as they did not engraft.

Supp Fig 14- is mislabeled Sup Fig 1. How many times performed? Why does BB have a stronger response to Il-7 than leukaemia? Is that expected?

Response

The mislabeling occurred in the conversion from word document to PDF. This was now fixed.

The analysis was done only for the leukemia and the BB cells with matching batch – hence only done once.

As for the response of BB cells to IL7, this is shown in Pro-BI stage- where normally cells have IL7RA expression and are expected to respond to IL7. The Leukemic cells are almost exclusively in Pro-BII stage, and the signal shown in this panel is mostly from non-Leukemic cells.

Supp Figure 15- is mislabeled Sup Fig 2. It should probably show RAG2 as well

Response

The mislabeling occurred in the conversion from word document to PDF.

Supp Figure 15 is now part of figure 4. RAG2 expression was added to the figure (see below)

REVIEWER COMMENTS

Reviewer #4 (Remarks to the Author):

Thank you for your detailed responses. I am satisfied that the revisions made add clarity to the manuscript. I have no further suggestions.